



# Evaluation of the WRF model with different domain configurations and spin-up time in reproducing a sub-daily extreme rainfall event in Beijing, China

Qi Chu[1,2], Zongxue Xu[1], Yiheng Chen[2], and Dawei Han[2]

[1] College of Water Sciences, Beijing Normal University, Beijing, 100085, China

[2] Department of Civil Engineering, University of Bristol, Bristol, BS8 1TR, UK

*Correspondence to*: Zongxue Xu (zongxuexu@vip.sina.com)

**Abstract.** The use of rainfall outputs from the latest convection-scale Weather Research and Forecasting (WRF) model is proven to be an effective way to extend the prediction lead time for flood forecasting. In this study, the effects of WRF domain configurations and spin-up time on rainfall simulations were evaluated at high temporal (sub-daily) and spatial (convective-permitting) scales for simulating a regional sub-daily extreme rainfall event occurred in Beijing, China. Seven objective verification metrics calculated against the ground precipitation observations and the ERA-Interim reanalysis, were analyzed jointly by the subjective verification to explore the likely best set of domain configurations and spin-up time. It was found that the rainfall simulations were quite sensitive to the change of the WRF domain size and spin-up time when evaluated at the convective scale. A model run with 1:5:5 horizontal downscaling ratio (1.6 km), 57 vertical layers (0.5 km), and 60-hour spin-up time covering Northern China exhibited the best skill in terms of the accuracy of rainfall intensity and the spatial correlation coefficient (R). Comparison made between the optimal run with the above set of the configurations and the initial run of the comparative test setup based on the most common settings revealed an evidential increase in each verification metric after the evaluation process, with R increased from 0.49 to 0.678, the relative error of point maximum precipitation rose from 0.41 to 0.881, and the spatial accumulated error fell by 43.22 %. In summary, the reevaluation of the domain configurations and spin-up time is of great importance and worthwhile in improving the accuracy and reliability of the rainfall simulations in the regional sub-daily heavy rainfall (SDHR) applications.





**1 Introduction**
The potential for the sub-daily heavy rainfall (SDHR) to increase with climate change is of significant societal concern, with
SDHR-derived flash floods (FF) being one of the most destructive natural hazards threatening many urban areas in Northern
and Central China, as well as many other parts of the world. In these regions, SDHR is mainly triggered by the Regional
Mesoscale Circulation Systems (MCSs) and detected with increased intensity and frequency in warm seasons (Yu et al., 2007;
Chen et al., 2013). Records from the Emergency Events Database (EM-DAT) indicate that the damages and losses caused by
the FF events in China have increased significantly over the past decades. And the risks of such hazards are expected to
increase significantly due to the continuing positive trend predicted in SDHR's magnitude by most General Circulation
models (Chen et al., 2012; Willems et al., 2012; Westra et al., 2014). Another noteworthy factor to the upward risk is the
accelerating urbanization, which has already changed the hydrologic characteristics of the land surfaces considerably,
resulting in higher peak flows and shorter flow concentration times (Xu and Zhao, 2016; Gao et al., 2017). In such cases,
very short-term (0-6 hours) rainfall predications are not sufficient to provide adequate warning and mobilizing emergency
response, particularly over the medium or large urban areas with decreased hydrologic response time (Shih et al., 2014; Li et
al., 2017). Therefore, Numerical Weather Prediction (NWP) models for rainfall predictions with longer lead time have
become more popular in flood-related studies and applications (Cuo et al., 2011).
Precipitation uncertainty accounts for a large proportion in the uncertainty of flood forecasts. Hence, NWP had long been
questioned for its use in flood forecasting due to the clear uncertainty of itself (Castelli, 1995; Bartholmes and Todini, 2005).
The ice was broken upon considerable improvements have been achieved in its predictive skill since 2000, when high
resolution data assimilation and physical parameterizations suitable for convection-scale NWP modeling were enabled with
demonstrated reliability (Done et al., 2004; Clark et al., 2016). Experimental studies suggest that a nested,
convective-permitting NWP model can capture more fine-scale ingredients and triggers of the convective storms (Klemp,
2006; Prein et al., 2015). Until now, most NWP models adopted by the meteorological services and research communities
allow convective-scale modeling, including the latest Weather Research and Forecasting (WRF) Model. However, limited by
current computational power and storage capacity, they can only supply the global-scale rainfall predictions up to 10
kilometers (WMO, 2014), which is still quite coarse for regional FF forecasting. To deal with this issue, more researchers
begin to carry out high-resolution regional case studies by downscaling the global NWP products to the domains of interest
(Hong and Lee, 2009; Soares et al., 2012; Sikder and Hossain, 2016). Results from these studies show that at a shorter time
range, a high-resolution regional NWP modeling (e.g. WRF) can produce better weather forecasts than the global one, as it



can better resolve surface heterogeneity, topography and the small-scale features in the flow, including growing instabilities
(Miguez et al., 2004; En-Tao et al., 2010; Prein et al., 2015).
Despite the potential of NWP for forecasting heavy rainfall, the prediction of NWP still contains a lot of uncertainties arising
from initial and boundary conditions and uncertainty in model physics, and they can be exaggerated by the chaotic nature of
NWP system. In a limited-area modeling, the effects caused by those factors are expected to be magnified during the
downscaling procedure and the model settings are supposed to be reevaluated and calibrated (Warner, 2011; Vrac et al, 2012;
Liu et al., 2012). Taking WRF model for example, being modeled in a high-resolution scale means that the convective
processes is more likely resolved by explicit physical schemes than the coarser run with implicit solutions, and may
incorporate new uncertain info from the model physics (Done et al., 2004; Ruiz et al., 2010; Créat et al., 2012). Except for
the model physics, studies suggest that some model configurations can also have a visible effect on the rainfall forecasts on a
watershed scale by influencing the initial and boundary conditions (Aligo et al., 2009; Fierro, 2009; Cuo et al., 2011).
However, those model configurations have been received much less attention in the high-resolution case studies due to their
relatively insignificant effect on rainfall forecasts compared to the effect of the model physics on the coarser-scale and
long-time run. For the computational facility, these model configurations, such as domain configurations and spin-up time,
are often left with the familiar regional settings recommended by the official website of WRF model and some experimental
heavy rainfall studies.
Given that precipitation is the most sensitive variable to the model uncertainties, in this study, reevaluation is implemented
on WRF to investigate whether the recommended WRF configurations are the best choices in reproducing a regional SDHR
event. WRF model is adopted here for assessment because of its superior scalability and compute efficiency which are
valued in interdisciplinary studies (Klemp, 2006; Foley et al., 2012; Coen et al., 2013; Yucel et al., 2015). As the latest NWP
community model, WRF has provided current developments in physics, numerics and data assimilation, and thus become
widely used in both theoretical studies and practical applications (Powers et al., 2017). The SDHR event selected here is a
sub-daily heavy rainfall occurred on July 21st, 2012 in Beijing, China. Beijing is one of the most vulnerable cities to
SDHR-induced floods in Central China (Yu et al., 2007). Precipitation in this area is mainly caused by monsoon weather
systems and enhanced by the local orographic effects, with 60 % - 80 % precipitation concentrating on a few SDHR events
during warm seasons (Xu and Chu, 2015). The rainfall happened on 21 July 2012 is the most disastrous urban flood event
with the largest precipitation since 1961, resulting in 79 deaths and more than 1.6 billion dollars in damage due to the failure
of the operational NWP system to predict it (Wang et al., 2013; Zhou et al., 2013). As such, some convective-scale studies
based on ARW-WRF are carried out to reevaluate the optimal model physics (Di et al., 2015; Wang et al., 2015), which have
provided a favorable background to stimulate and implement this research.





The second question we attempt to explore is that, if the recommended model configurations are not the best choices, then to
what extent the rainfall forecasts could be improved with the likely best set of settings. Model configurations selected here
for evaluation are domain size, vertical resolution, horizontal resolution and spin-up time that are demonstrated influential on
daily-scale extreme rainfall forecasts (Leduc and Laprise, 2009; Aligo et al., 2009; Goswami et al., 2012). Comparative test
with four scenarios is designed with each scenario focused on one configuration to ensure forecasting disparities being
attributed solely to one factor each time. The entire test is conceived as a progressive process to help quantify the overall
improvement in the accuracy of simulated rainfall. That is to say, the optimal setting evaluated in the former scenario will be
adopted as the primary choice for the next scenario. The dataset used for 'ground truth' is a grid observation dataset from
China Meteorological Center. A coaster-scale reanalysis called ERA-Interim is also utilized to monitor the possible departure
of the simulations from the driving weather fields associated with the variations of the settings. Since no single verification
approach is currently demonstrated capable of yielding complete information about the quality of rainfall predictions (Sikder
and Hossain, 2016), seven objective verification metrics depicting different features of the rainfall performance are adopted
and viewed jointly using the subjective verification. Most metrics adopted in this study are the metrics used for assessing the
performance of WRF in simulating heavy rainfall with daily or longer time periods (Liu et al., 2012). Here, they are
calculated hourly and averaged in different sub-daily time span to evaluate the effect of the WRF domain configuration and
spin-up time from a sub-daily and convective-scale perspective.
**2 Numerical Models for Heavy Rainfall Forecasting**
The dataset used for WRF downscaling was a global atmospheric reanalysis called ERA-Interim. It is produced with
Integrated Forecasting system (IFS) developed by the European Centre for Medium-Range Weather Forecasts (ECMWF).
The IFS is an Earth-system system that incorporates an atmospheric model that is fully coupled with the land surface and
oceanic processes, and a data assimilation system. The atmospheric model is used to provide simulated observations per 30
min with a spectra resolution of T255 (approximately 81 km over Beijing). It is then served as the prior information and
combined with available observations twice a day using the four-dimensional variation (4D-Var) assimilated system to
produce the reanalysis. The final reanalysis product, ERA-Interim, currently contains 3-hourly estimates of a large amount of
the surface parameters, and 6-hourly gridded estimates of meteorological variables with spectra resolution of T255 on 60
vertical levels, for all the dates from 1 January 1989 (Berrisford et al., 2009; Dee et al., 2011).
The Advanced WRF (ARW-WRF), Version 3.7.2, was utilized to dynamically downscale the global ERA-Interim dataset.
ARW-WRF is a compressible non-hydrostatic and convective-permitting regional NWP model built on Dynamic Euler





Conservative-Form Equations. As the latest regional mesoscale NWP community system, WRF is featured with two
dynamic cores, a data assimilation system and a platform facilitating parallel computation and function portability. The data
used to initialize the model could be observations, model analysis data or assimilated reanalysis data. For model
discretization, it uses a third-order Runge-Kutta method for temporal separation and Arakawa-C grid staggering scheme for
spatial discretization. The model is capable of conducting either one-way or two-way nested run for regional downscaling.
Detailed introduction on the physics and numeric of WRF can be found in Skamarock and Klemp (2008). With an emphasis
on efficiency, portability and the state-of-the-art update from research to application, WRF has been widely incorporated in
various operational systems, such as the Hurricane-WRF system for hurricane forecast and the WRF-Hydro system for
hydrologic prediction.
When using WRF in local area, the artificially assigned domain size and grid spacing for each nesting domain can introduce
additional uncertainties and errors into the downscaled features. Domain size implicitly determines the large-scale dynamics
and terrain effects, while the vertical and horizontal grid spacing determines the smallest resolvable scale (Goswami et al.,
2012). They two together (here referred to domain configurations) affect the spectrum of resolved scale and the nature of
scale interaction in the model dynamics (Leduc and Laprise, 2009), therefore responsible for precipitation generation and
distributions. In a fine-scale simulation, a small domain is preferred for computational efficiency, and the simulation with a
small domain is more likely to benefit from the lateral boundary conditions (LBC) to dampen the feedback between a
perturbation from the surface forcing and the large-scale general circulation (Seth and Rojas, 2003). But a domain with too
small size is not sufficient to enable full development of small-scale features over the area of interest. To deal with this issue,
the official website of WRF has provided a general guidance (Warner, 2011), in which the range of domains is recommended
to involve the patterns of the leading MCS features and surface perturbations, with more than 10 grids between two adjacent
nested domains.
Theoretically, one would like to suppose that a run with finer grid spacing could resolve more small-scale phenomena of
interest that are not presented in LBC, and hence get more accurate forecasts. This is demonstrated true for WRF model when
comparing a coaster run (>10 km horizontally or >1 km vertically) with a convective-scale run (1~5 km horizontally or <1
km vertically) in representing the convective storms. However, when the comparison is conducted among the
convective-scale model runs, converse results are shown in different case studies. Taking horizontal resolution for example,
although evidences shows that simulations with higher resolution could capture more convective-scale features, the accuracy
of heavy rainfall forecasts is predicted with either great or no statistical improvements (Roberts and Lean, 2008; Kain et al.,
2008; Schwartz et al., 2009). The work of Fierro (2009) suggests that the features detected in the finer-resolution simulations
tend to weaken the kinetic structures that favor the torrential rainfall intensity. Similar conclusion is achieved by Aligo et al.



(2009) on evaluating the impact of WRF vertical grid spacing on summer rainfall simulations. Since then, approximately 4
km horizontal grid spacing and 1 km vertical grid spacing are employed in several related studies as a compromise choice, as
well as to save the load for calculation.
For regional weather forecasts, a spin-up period is often required to balance the inconsistencies between the physics in the
WRF model and those imposed by the initial conditions from ERA-Interim reanalysis (Luna et al., 2013). The proper spin-up
duration is dependent on the time needed for initialization, therefore it can be affected by the range of domain and the local
boundary perturbations (Warner, 1995; Kleczek et al., 2014). Besides, it is also limited by the chaotic nature of the NWP
system, resulting in deterioration of the predictive skill as time evolves. Hence, in real-time rainfall forecasting when a short
spin-up time is expected, the proper duration is affected by the domain size and regional boundary conditions. But for the
areas that with shorter hydrologic response time, a longer spin-up time is needed to allow sufficient time for warning and
motivating emergency response. To compensate the effects caused by the chaotic nature within a longer time span, lateral
boundary information is updated regularly by the latest forecasts or analysis to adjust the regional model simulations. In such
a case, the best-fit performance may occur in the run with longer spin-up time. Based on previous sensitivity studies, the
official WRF website recommends a spin-up time of 12 hour as the initial state, yet it is then supposed to be the most suitable
choice in many case studies without further verification.

## 3 Study Event and Experimental Design

As mentioned before, one aim of this study is to reevaluate whether the recommended domain configuration and spin-up
time of the WRF model is the best set of configurations when verified at a sub-daily scale in reproducing a regional SDHR
event. Here, the sub-daily heavy rainfall event occurred on 21 July 2012 in Beijing, China was selected as the case study.
Before introducing the entire experiment design procedure, the reason for choosing this event, the synoptic and physical
feature behind this event, and the model physics adopted in this study were firstly presented in the following section.

### 3.1 Study Event Selection and WRF configurations

In this study, Beijing was selected as the study area because it is one of the most vulnerable cities to SDHR-induced FF
hazards in China. Beijing is located in Central China with an area of 16 411 km$^2$, and its weather is mainly affected by the
semi-humid warm continental monsoon climate. The airflows favoring the local precipitation are the cold, dry airflow from
the northward high-latitude area and the hot, wet airflow from the southern oceans. The interaction of these two airflows in
different seasons leads to clear divergence in the temporal distribution of rainfall amount, with 60 %-80 % precipitation
concentrated on few heavy rain events during the warm seasons. Of all the heavy rainfall events, the intensity and frequency
of SDHR are detected with the greatest upward tendency over the past decades. Meanwhile, Beijing, as the capital of China,





has experienced significant urbanization expansion associated with rapid development in population and economics.
Negative feedbacks of the development, such as the loss of natural water bodies, the increase of low permeable land cover
and urban drainage pipe networks, lead to the continuing decreased hydrologic response time. Besides, the area where a large
amount of the population lives is in the southwestern plain area, which is the downstream of the mountain regions featured
with steep terrain varying from 2 300 m to 60 m (**Fig. 1**). These factors all contribute to the increased exposure of this city to
high flooding and waterlogging risks when SDHR happens (Xu and Chu, 2015).
**[Figure 1]**
The selected case study is the largest rainfall event occurred in Beijing from the past 65 years ago. The rain lasted for 16
hours from 2 am to 6 pm on 21 July 2012 (UTC), with highest hourly rainfall intensity (100 mm) hit the Southwest part of the
plain area. The derived FF hazard led to 79 deaths, 1.6 billion US dollars in damage and affected more than 1.6 million
people. In addition to Beijing, the adjacent provinces including Hubei and Liaoning were all significantly affected by the
heavy rainfall and experienced severe FF hazards. The synoptic features triggered the SDHR were the eastward-moving
vortex in the mid-high troposphere, the north-moving subtropical high pressure and the sharp vortices wind shear (Sun et al.,
2013). The whole physical process of this event could be divided into two phases. From 2 am to 2 pm, the convective rain
was dominated and enhanced by the orographic effect. The frontal rain then superseded after the arriving of the cold front
moving from the northwest till 6 pm (Guo et al., 2015). Compared to the first phase, the intensity of precipitation in the
second phase was relatively lower than the first stage as there was less strong kinetic forcing to maintain the precipitation
process.
In this study, the ERA-Interim Reanalysis and 30-second static geographical data were employed to initialize the surface and
meteorological fields of WRF. As shown in **Fig. 2**, ERA-Interim captures the vortex and the subtropical high pressure well at
the beginning of the rain. Besides, the pattern of the MCSs and the primary synoptic features shown in this figure also
corresponds well to those described in the previous literature (Zhou et al., 2013). Regarding the setup of model physics, the
configurations adopted in this study were mainly based on the results of the high-resolution sensitive studies of the WRF
model physics in simulating the same event (Wang et al., 2015; Di et al., 2015). Therein, 'Resolved rain' and 'convective rain'
were driven by the single moment 6-class microphysics scheme (Hong and Lim., 2006) and the Grell-Devenyi cumulus
parameterization scheme (Grell and Devenyi, 2002), respectively. The Noah land surface model (Chen and Dudhia, 2001)
was adopted as the land surface scheme, coupled with the surface layer model once utilized in MM5 (Ek et al., 2003). The
radiation processes were represented by the RRTM short-wave radiation (Mlawer et al., 1998) and the RRTM long-wave



radiation schemes (Mlawer et al., 1997), respectively. As for the planetary boundary layer scheme, the Yonsei University
planetary method (Hong et al., 2006) was used.

4                                     **[Figure 2]**

**3.2 Experimental Design on Domain Configurations and Spin-Up Time**
As mentioned before, the comparative test was designed as a progressive process to help quantify the overall improvement in
the predictive skill of the WRF model for simulating the Beijing SDHR event after reevaluating the WRF domain
configurations and spin-up time experiments. The comparative test was classified into four successive scenarios: the first
three relate to the domain settings, including domain size, vertical resolution, and horizontal resolution, followed by the one
concerning spin-up time. During the whole procedure, the optimum configuration assessed in the former scenario was then
adopted as the primary choice for the corresponding setting in the next scenario. For facilitating comparison, the initial fields
and the model physics mentioned in the last section were used and kept the same throughout the entire comparative
procedure. Besides, the Lambert conformal horizontal projection centered at $42.25\,^{\circ}$ N, $114.0\,^{\circ}$ E, and the sigma vertical
coordinate with the top level of 50 hpa were applied, along with the automatic choice of the integral time step to balance the
conflict between computational efficiency and numerical stability.
Before starting the test, the initial state of WRF domain configurations and spin-up time were set up based on the
recommended choices described in Section 2. To ensure the horizontal resolution in the smallest domain could be highly
enough to explicitly resolve the convection-scale processes, three levels of nested domains were adopted (**Fig. 1**). The
outermost domain (D01) had 40.5 km horizontal grid spacing and covered Northern-Central China where the main perturbed
synoptic features were involved. To reduce the initial error introduced by interpolating the initial fields to the assigned
domains, the ranges of the domains were all set up along certain delineation grid-line of the ERA-Interim dataset, as well as
the adoption of an odd horizontal downscaling ratio (1:3:3). The grid spacing of the innermost domain (D03) was nearly 4.5
km with the domain covering the entire Beijing area. The second domain (D02) was the child of D01 and the parent of D03.
The distance between D01 and D02 was similar to that between D02 and D03, both of which were more than ten grids
distance. Here, two-way nesting scheme was used to balance the inconsistency between the model physics of the inner
domain and the LBC information forced from the outer domain. Since the vertical grid spacing of ERA-Interim reanalysis is
around 1 km, the same vertical levels were utilized in the initial run. 12 hours (12 h) spin-up time was selected with the
output saved every 1 hour (1 h), and the LBC was updated every 6 hours (6 h) by the use of ERA-Interim.





As shown in **Table 1**, the first experiment (C0) was set up based on the model configurations mentioned above. To test if the
domain configuration and spin-up time adopted in C0 was the likely best set of those configurations, four scenarios were
designed. The first scenario (S1) was focused on evaluating the effect of the horizontal domain size. For computational
efficiency, the leading MCSs systems driven the synoptic features was not completely involved within the outermost domain
of C0, whose info was compensated by the updated LBC from ERA-Interim. To verify whether the assigned domain size in
C0 was large enough to enable full development of small-scale features, two comparative experiments of C1 and C2 were
devised. The outermost domain (D01) size of C2 was the largest, which incorporated the leading MCSs systems and covered
the entire Northeastern Hemisphere. The intermediary domain (D02) of C2, centered between the outermost domain and the
innermost domain (D03), was then adopted as the outermost domain of C1. The second scenario (S2) consisted of three
experiments with decreased vertical grid spacing or increased vertical levels. Here, the setup of the starting experiment in S2
was equal to the settings of the optimal case evaluated in S1 (OS1), and the same rule was followed in the third (S3) and the
fourth scenario (S4). In S2, the vertical levels of C3 and C4 were one and two times more than that of OS1, respectively, to
test if the model run with finer vertical resolution could yield improved performance in rainfall simulations. S3 assessed the
effect of the horizontal resolution, with increased downscaling ratios of 1:5:5 in C5 and 1:7:7 in C6. S4 was composed of 13
experiments (C8-C18), except C7 having no warming time, with spin-up time increasing by 12 hours (12 h) from C8 (run 24
hours ahead), to search for the reasonable longest spin-up time to well reproduce the SDHR event occurred in Beijing.

18                                           **[Table 1]**

**4 Verification Schemes**
The short-duration rainfall intensity and location of the heavy rain-belt are of great concern for urban flash flood mitigation,
and the accuracy of rainfall simulations at each grid are equally important concerning the potential risks of waterlogging and
flooding. Therefore, in this study, the temporal step selected for verification was an hour, and the smallest spatial scale for
calibration was equal to the grid size of the model outputs. In WRF, two-way nesting scheme updates the parent domain by
using the results of the child domain. Thus, the intermediary domain (D02) that covered a large portion of the rainfall
forming in this severe SDHR event was chosen as the analyzed range to compare the performance of the WRF domain
configuration and spin-up experiments. For reference, two grid datasets were used. One was an hourly 0.1-degree grid
rainfall data from the China Meteorological Center (CMC), produced by fusing the observed information from ground
stations and TRMM satellite data. The other one was the ERA-Interim reanalysis, utilized to monitor the departure between
the simulations of the model run with the coarser-scale boundary information provided by the ERA-Interim reanalysis.





Given that the sub-daily scale rainfall data was not available from the ERA-Interim, the atmospheric precipitable water vapor
(PW), which determines the possible maximum precipitation, was used instead to be compared with the same field of the
model outputs every 6 hours.
Two verification schemes, objective verification and subjective verification were used jointly to identify the optimal set of
WRF domain configurations and spin-up time. The selected seven objective verification metrics described different
characteristics of the rainfall simulations. Of all the metrics, five were rainfall-related calculated between the simulations and
the CMC grid observations, and the other two were relevant to PW by comparing the model outputs with the ERA-Interim
reanalysis. During the whole Beijing SDHR process, the characteristic of the precipitation varied at different time duration,
with little rain in the first 2 hours, heavy convective rain from 2 am to 2 pm, then turning into frontal rain till 6 pm. To explore
if the evaluated result would differ when the assessments were conducted within different time periods, four statistical time
durations (6 h, 12 h, 18 h, and 24 h) were utilized, counting from 12 am on 21 July 2012 (UTC).
For the verification, some of the metrics proposed by Liu et al. (2012) was used to evaluate the predictive skill of WRF by
comparing the model outputs with the CMC observations. The accumulated areal rainfall was assessed by the relative error
($RE_{TP}$). The categorical verification metric was chosen as the probability of detection (POD), which indicates the percentage
of correct simulated rainfall hits. The continuous metric was selected as the root mean square error (RMSE) that shows the
amount of error in the predicted precipitation. Besides those three metrics, the Pearson correlation coefficients (R) that
describes the spatial association of the simulations and rainfall observations, and the relative error for the maximum areal
precipitation ($RE_{PMAX}$) were used as well (**Eq. (1)** and **Eq. (2)**). Since the evaluated temporal dimension (no more than a day)
was much less than the spatial dimension (the number of grids inside the assigned domain), all the metrics were firstly
calculated at the spatial dimension. That is, the metrics were firstly computed between the observations of each grid and the
simulations of the same grid at each saved time step (1 h), and then averaged within different time durations (6 h, 12 h, 18 h,
or 24 h) for final analysis.
As for the evaluation of PW, the root mean square error (WRMSE) and the Pearson correlation coefficients (WR) were
chosen to measure the departure between the simulations of the model run and boundary information provided by
ERA-Interim. Besides, to facilitate evaluation, all the metrics were rescaled to have the ideal score of 1. The correlations
between the original value and the rescaled value of the metrics were shown in **Table 2**. As each metric describes different
features of the rainfall simulations, the values of these indices were checked and viewed jointly by subject verification to find
out the likely best set of the domain configurations, and search for the reasonable longest spin-up time.





$$R = \frac{1}{N}\sum_{i=1}^{N}\left(\frac{\sum_{j=1}^{M}(f_j-\bar{f})(r_j-\bar{r})}{\sqrt{\sum_{j=1}^{M}(f_j-\bar{f})^2 \sum_{j=1}^{M}(r_j-\bar{r})^2}}\right) \qquad (1)$$
$$RE = \frac{1}{N}\sum_{i=1}^{N}\left[\frac{f-r}{r}\times 100\%\right] \qquad (2)$$
where $R$ is the empirical spatial correlation coefficient, $M$ is the total number of grids covering the D02 of the initial
experiment in each scenario. $f_j$ is the value of the $j$ th grid of the tested field at the time step $i$, and $r_j$ is the value of the
referenced field. $N$ is the total number of time steps, which is 6, 12, 18, or 24 when evaluated within different time periods.
RE is the relative error. For the maximum areal precipitation, $f$ is the tested value of the maximum grid precipitation over the
area of interest. $r$ is the referenced value of CMC observations, which is calculated by the inverse distance square method
based on the model simulations.
**[Table 2]**
**5 Results and Analyses**
The WRF experiments of the first three scenarios were run from 12 pm on 20 July 2012 to 12 am on 22 July 2012 to identify
the likely best set of WRF domain configurations. Corresponding to the sequence of the scenarios listed in **Table 1**, the
evaluation of WRF domain size scenario (S1) was first presented. **Fig. 3** shows the spatial values of the verification metrics
of the domain size experiments calculated within 6 h, 12 h, 18 h and 24 h time periods counted from 12 am on 21 July 2012.
The experiment with the likely best performance in S1 was chosen as the starting experiment in the next vertical resolution
scenario (S2) with 29 vertical levels set up based on the grid spacing of the ERA-Interim reanalysis. The other two
experiments were designed with doubled and tripled vertical levels than OS1 to evaluate the effect of vertical resolutions on
the rainfall simulations. Similar to **Fig. 3**, the performance of the WRF vertical resolution experiments was present in **Fig. 4**.
The third horizontal resolution scenario (S3) also selected the best-performed experiment in S2 (OS2) as its initial
experiment with decreasing horizontal downscaling ratio of 1:5:5 in C5 and 1:7:7 in C6. The calculated values of the metrics
averaged within 6 h, 12 h, 18 h, and 24 h time periods were shown in **Fig. 5**.
**5.1 Results of the Domain Size Scenario**
It is clear that the predictive skill of the WRF domain size experiments for the rainfall is reduced as the calculated temporal
duration increases from 6 h to 24 h (**Fig. 3**). The most evidential deteriorations are detected in the point-to-point accuracy of
the rainfall, with the percentage of the correct rainfall hits (POD) decreased by 5%, and two times increase in the spatial
accumulated errors. The spatial association between the simulations and the CMC observations declines as well, with the





correlation coefficient (R) decreased by 0.2 on average. The relative bias for the accumulated areal rainfall reveals that the
WRF domain size experiments tend to overestimate the total rainfall amount throughout the entire simulated process. As for
the point maximum precipitation, although being overestimated in the first duration (6 h), an obvious negative bias exhibits
when the heavy convective rain begins. When the simulations were compared with the ERA-Interim reanalysis, slight
growth was found in the discrepancy between the model outputs and the forced PW fields (WRMSE), while a significant
increase was detected in the spatial correlation coefficient for PW (WR) as the time duration increases from 6 h to 24 h. This
may be due to the role of the updated boundary info in adjusting the simulations of the long model run to approach the
large-scale general circulation conditions.

10                                                 **[Figure 3]**

The range of the intermediary domain (D02) is different in the domain size scenario, to facilitate comparison, the smallest
domain size of D02 in C0 that involving the main synoptic features triggered the SDHR event was adopted as the initial
analyzed range. By comparing the four subfigures in **Fig. 3**, it is found that the values of all the metrics within a given period
do not point to one perfect experiment, and their rank in the performance showed by a certain metric differs when the
averaged time periods changed. At the early stage of the rain (6 h), the performance of C1 is relatively superior to the other
two experiments regarding the accuracy and association of the precipitation amount, and the superiority remains when heavy
convective rain dominates. When coming to the end of the rainfall, although the spatial correlation and the maximum
precipitation of C1 are still the highest, the accuracy of the rainfall simulations decreases significantly, and more exact point
rainfall amount and the percentage of correct hits are detected in C0. As we have mentioned before, a small domain size is
more likely to be influenced by the boundary information, and this is demonstrated true when comparing the values of
WRMSE and WR, with C0 having the highest similarity and C2 having the lowest the similarity with the PW fields of the
reanalysis through most of the time. However, it is noteworthy that either at the first stage (6 h) or in the heavy rain period (12
h, 18 h), the maximum precipitation predicted by C0 is much lower than that of C1 and C2, and the spatial correlation of the
rainfall in C0 was obviously less than that of C1. This indicates that the size of C0 is not broad enough to allow full
development of the small features by the physics of model run. Besides, C2 with the largest domain range is not showing
better performance than C1. Based on the comparison between the WRMSE and WR of C1 and C2, the reason could be
attributed to the inefficient use of the boundary conditions in C2 to adjust the false perturbations arising from the local model
run. In summary, C1 is verified with the best performance in this scenario.



**5.2 Results of the Vertical Resolution Scenario**
Since C1 is selected as the optimal experiment (OS1) in the WRF vertical resolution scenario (S2), the analysis was then
made on the entire intermediary domain of C1 that covers the large portion of the rain-belt formed in the heavy rainfall event.
Due to this change, fairly higher values of the metrics of C1 are shown in **Fig. 4**. By comparing the values of WRMSE
calculated under different domain size, it is clear that the updated boundary conditions more likely influence the simulations
in the expanded portion of the analyzed area. This could explain why the accuracy of the rainfall increases as well as
simulates with higher POD and RMSE values during the whole rainfall process. For all the experiments in S2, the significant
downward trend is also found in capturing the correct hits and point rainfall amount with the increase of the calculated
temporal duration. The temporal variation tendency of TP and PMAX detected in S2 are similar to the trend shown in S1.
The difference is that the R between the simulations and CMC observations varied a little and almost kept the same within
different time periods. Besides, the performance of the S2 experiments tends to be less sensitivity to the boundary conditions
that shows little variation in WRMSE and WR.

14                                **[Figure 4]**

Compared with the apparent discrepancy shown in the metrics of the domain size experiments, the differences in the rainfall
metrics between the experiments with different vertical levels are not that obvious, especially in the less rainy period (6 h)
and the period when the convective rainfall dominated (12 h). Across all the conditions, C3 shows a better agreement with
the CMC observations than the other two experiments regarding the accurate and spatial correlation of the rainfall amount.
By comparing C3 and C1, we can see that a finer vertical resolution may increase the possibility of WRF to explicitly resolve
the small-scale model physics and enhance the accuracy of the rainfall amount and the rainfall locations. However, it is clear
that C3 predicts less maximum precipitation than C1. This may be because a finer vertical resolution can magnify the
propagation of the surface perturbation across the vertical grid columns that may weaken the kinetic energy that favors
precipitation. The comparison made between C3 and C4 shows that although with further refinement of the vertical
resolution, the performance of C4 is worse than that of C3. By analyzing the values of WRMSE and WR, it shows that the
departure from the simulations to the reanalysis is more distinct in C4. This discrepancy may arise from the exaggeration of
the initial errors during the interpolation process and the incorporation of the false surface perturbations limited by the
accuracy and resolution of the initial forced data. Given the above analysis, C3 is then considered to be with the best
performance as it predicts more accurate rainfall simulations in the heavy rainy periods.





**5.3 Results of the Horizontal Resolution**
Based on the evaluated results in the S2 scenario, OS2 is therefore set up as C3 in the horizontal resolution scenario (S3). The
predictive skill of the S3 experiments is detected with the similar temporal tendency to that of the S2 experiments (**Fig. 4** and
**Fig. 5**). However, the sensitivity of the metrics to the variation of the horizontal resolution is more evident than that with
different vertical resolutions. During most time periods, C5 was detected with better performance than C3 and C6. The
comparison made between C3 and C5 shows that C5 tends to predict more accurate spatial patterns of rainfall across the
entire rainfall process. The result of WRSME indicates that model run with higher horizontal resolution seems to be more
likely benefited from the updated lateral boundary conditions. Higher values of PMAX and TP are also detected in C5
against C3. This may contribute to the explicit resolving of the model physics. Then it could explain why the performance of
C6 is better than C5 in the first 6 h periods. Noteworthy is that the predictive skill of C6 deteriorates rapidly since the heavy
rainy period (12 h, 18 h) with the lowest POD, RMSE, and R till the end. Analysis of the WRMSE suggests that the
simulations of model run departures significantly from the coarser-scale PW fields used to force the model physics.
Theoretically, this may be due to the accumulated errors arisen from the imperfect model physics, or the initial and boundary
conditions that could be exaggerated by the chaotic nature of the NWP system. According to the above analysis, C5 is
deduced with the best predictive skill of all the S3 experiments.
**[Figure 5]**
**5.4 Searching for the likely ideal spin-up time**
The evaluation of the WRF spin-up time scenario (S4) is placed at the end of the experiment design, after reducing the
possible errors induced by the inappropriate domain configurations, so as to lower the influence of NWP's chaotic nature on
the model simulations and extend the forecasting time. In S4, C5 is adopted as the initial experiment (OS3). Unlike the
previous scenarios, the rank of the spin-up time experiments sorted by the metrics nearly stays the same across different
statistical time periods. As such, **Fig. 6** only presents the performance of the S4 experiments within 18 h time periods. We can
see that the predictive skill of WRF for the rainfall varies significantly to the variation of spin-up time. For most metrics, an
obvious downward tendency is found from 0 h to 36 h, and a short-term growth follows till 60 h and then with random
fluctuations after 72 h. Before 72 h, the variation of the rainfall metrics and PW metrics are almost consistent, which means
the good-fit simulations that with longer spin-up time is also physically reasonable during this period. The difference among
these experiments may be due to the different initial weather conditions (e.g. the water vapor amount and the started running
time of day). From the TP, it is found that all the spin-up time experiments overestimate the total rainfall amount. Regarding
PMAX, positive biases are detected in the simulations with 24 h and 48 h spin-up time, the time when the amount of water





vapor is the largest. Of all the metrics, POD is found with the least sensitivity to the spin-up time, but it has similar variation
tendency with TP, R, RMSE, and WR before 72 h, with the highest value occurred in 60 h (C11). Although relative different
change patterns are presented in PMAX and WRMSE, the best agreement with the observations is still detected in 60 h (C11).
Overall, C11 is regarded as the best experiment with the optimal set of domain configurations and spin-up time in
reproducing this SDHR event.

7                                                          **[Figure 6]**

**6 Discussions**
Previous analyses showed that the experiment with the default combination of those configurations (C0) was not the one
showing the best fit with the grid-based rainfall observations. In the domain size scenario, C0 was detected with too small
size to allow full development of the small-scale features and resulted in poor performance at the early stage of the rain (6 h,
12 h). The further refinement of C0's grid spacing in S2 and S3 demonstrated to be effective in enabling more explicitly
resolution and able to capture the spatial pattern of the rainfall better. The comparison made in the S4 scenario suggested that
the choice of the proper spin-up time was not only determined by the initialization time but also affected by the initial
weather conditions fed to drive the model. Meanwhile, the results also revealed that the experiment with too large domain
size, too high spatial resolution, or too long spin-up time could also get poor performance in rainfall simulations. Therefore,
the reasonability of these WRF settings should be checked before being utilized in the regional NWP systems for flood
forecasting or as the reference for flood mitigation design.
Besides exploring whether the recommended combination of WRF domain configurations and spin-up time was the best and
reasonable choice when used in the regional SDHR areas, the improvement in the performance was also evaluated by
comparing the spatial values of the verification metrics among the experiments. The values of the experiments assessed with
optimal performance in each scenario were listed in **Table 3,** to be compared with those calculated in the most initial
experiment (C0). Here, the 18 h time periods were selected for evaluation as it covered the entire heavy rain process and the
metrics calculated in this period showed better spectrum in identifying the best performance experiment than those computed
within the 24 h time periods. One exception was the domain size scenario, in which C0 had worse performance at the early
stage of the rain. Therefore, the improvement of C1 against C0 was mainly represented by the R and PMAX in the 18 h time
periods. Two lines of data were shown in C1, with each line of values calculated over different domain areas. It is clear that
the model outputs in the expanded area benefited more from the lateral boundary conditions, and the overall simulations



remained the advantage in capturing the small-scale features across the initial analyzed region. The improvement made by
the refinement of vertical resolution mainly represented in RMSE and R, but with the decreasing PMAX which may be due
to the weakened kinetic energy for favoring the rainfall. Higher values of POD, RMSE, R, and PMAX were detected in C5
when compared with C3, indicating that a proper increase in the horizontal resolution could increase the accuracy of rainfall
simulations. The largest difference of the metrics between C5 and C11 occurred in the PMAX, which may relate to the
difference in the initial weather conditions at different starting time of the run.

8                                                        [Table 3]

Overall, although the increased magnitudes were different among those metrics for the rainfall, they all exhibited an
enhancement tendency in the model predictability after the entire reevaluation process, with R increased from 0.49 in C0 to
0.678 in C11, RMSE (rescaled value, see **Table 2**) rose from 0.171 to 0.529, and PMAX rose from 0.41 to 0.881. As the
complete assessment is based on the objective verification metrics and checked by the subjective verification, it could
conclude that the domain configurations and spin-up time may have a significant influence in the regional sub-daily heavy
rainfall simulations. Therefore, it is certainly worth reevaluating those settings in the high-resolution regional studies, and
the accuracy of the heavy rain predictions could benefit obviously from these analyses. As for the evaluated metrics, it is
clear that the evaluation based on the metrics within one type or within one time period could achieve partial conclusions.
The use of multi-source dataset for verification can help analyze as comprehensively as possible, such as the use of WRMSE
and WR in this study. The use of different time durations could help better determine the physical reasonable optimal
configuration, such as the choice of the proper domain size. Of course, the verification results may also differ when the
interested fields and the interested tempo-spatial scales vary. To further understand the effect of those WRF model
configurations on the regional sub-daily heavy rainfall simulations, more objective-based verification metrics for SDHR
should be studied, and more SDHR evaluated case studies are needed as well.
**7 Conclusions**
In this study, the global ERA-Interim reanalysis was fed into the ARW-WRF model as the initial and boundary conditions to
simulate a sub-daily extreme rainfall event in Beijing, China. A progressive comparative test was designed to evaluate the
effect of the domain configurations and spin-up time on the ability of WRF to reproduce this extreme precipitation episode
by comparing the model outputs with the reference datasets involving the grid-based rainfall observations and the
ERA-Interim reanalysis. Five error metrics that describe different rainfall characteristics and two PW-related indices
monitoring the departure of model simulations from the coarser-scale reanalysis were grid-calculated within different
sub-daily time span. They were then checked and viewed jointly by subjective verification to pinpoint the likely best set of





the domain configurations and spin-up time and help quantify the possible improvement in the performance of WRF for
reproducing this severe sub-daily heavy rainfall event (SDHR) after implementing the entire reevaluation processes.
It is found that the precipitation simulations are sensitive to the change of the domain size, vertical grid resolution, horizontal
grid spacing and spin-up time. Of all the configurations, the most obvious variations are found when adjusting the domain
size and spin-up time of WRF. The analysis shows that the domain size merely covering the area of interest may not be broad
enough to allow full development of the small-scale features, resulting in poor performance in capturing the spatial pattern of
heavy rainfall especially in the early stage of the rain. Despite the dominant role of the chaotic nature, there is still a
possibility that the model run with a longer spin-up time could result in better rainfall simulations if with the favorable initial
weather conditions. Vertical resolution and horizontal resolution though show less impact, yet the accuracy of the rainfall
amount and correct hits exhibit evident increases if run with slightly higher spatial resolutions. By comparing the C11
evaluated with the optimum configurations and the C0 with the recommended settings, an apparent increase is detected in the
metrics, with R increases from 0.49 to 0.678, PMAX increases from 0.41 to 0.881, and the spatial accumulated error fell by
43.22 %. This therefore indicates the benefits of reevaluating the WRF domain configurations and spin-up time for the
regional sub-daily heavy rainfall studies.
With the intensified SDHR and the increased risks of the SDHR-induced hazards, more demand is raised from the
operational flood management communities for more accurate rainfall predictions with longer lead time beyond the
hydrologic response times in the highly affected areas. Up to now, the only way demonstrated effectively is to use the freely
available global NWP products and the high-resolution regional NWP model (like WRF) downscaling the fields to the area
of interest. Therefore, the uncertainty associated with the downscaling process should be well evaluated to ensure the
reliability of the produced rainfall before being utilized in flood forecasting systems. This study suggests the importance of
WRF domain configurations and the spin-up time in influencing the regional rainfall simulations and demonstrates large
improvement by reevaluating those settings. The metrics used here indicate that evaluation based on one category metric
only or the metrics within one time periods (e.g. 24 h) could not make a comprehensive comparison and may lead to partial
conclusions. To handle the conflict shown in the rainfall-related metrics, the use of PW fields that calculated against the
reanalysis demonstrated helpful to determine the optimal set of the domain configurations and spin-up time. Here, subjective
verification is used to check the physical reasonability of the objective metrics and help to make the final decision, so the
evaluation is still partly subjective-dependent. To simplify the assessment procedure, the verification methods served as a
replacement of the subjective verification is worthy exploring. More regional case studies are needed as well to further
investigate the effect of those configurations for simulating the regional SDHR. Besides, studies should also be carried out in
exploring the methods to reduce the uncertainties of the regional NWP models associated with the downscaling procedures.





**Competing interests**
The authors declare that they have no conflict of interest.
**Acknowledgement**
This study is supported by the key research project "Urban flood/waterlogging hazard and disaster reduction strategies in
Beijing" (8141003) of Beijing Natural Science Foundation. Support is also received from the Resilient Economy and Society
by Integrated Systems modeling (RESIST), Newton Fund via Natural Environment Research Council (NERC) and
Economic and Social Research Council (ESRC) (NE/N012143/1), and the National Natural Science Foundation of China
(No:4151101234). The China Scholarship Council supports the first author for her academic visit to University of Bristol,
UK.

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



**Figure captions**

**Figure 1: Relative location and the geometry feature of the study area.**

**Figure 2: Initial wind field and geopotential height field provided by the ERA-Interim reanalysis within D01 of C2 at 12 pm on 20 July 2012.**

**Figure 3: Spatial values of the verification metrics for the WRF domain size experiments in S1 calculated within different temporal durations.**

**Figure 4: As in Fig. 3, but for the experiments in S2 with different vertical resolution.**

**Figure 5: As in Fig. 3, but for the experiments in S3 with different horizontal resolution.**

**Figure 6: Spatial values of the verification metrics for the WRF spin-up experiments in S4 calculated within 18 h time periods.**

**Table captions**

**Table 1: Category of the experiments with different domain size, vertical resolution, horizontal resolution and spin-up time.**

**Table 2: Correlations between the original value and the rescaled value of the verification metrics.**

**Table 3: Comparison of the metrics (18 h) between the initial experiment and the evaluated optimum experiments in each scenario.**





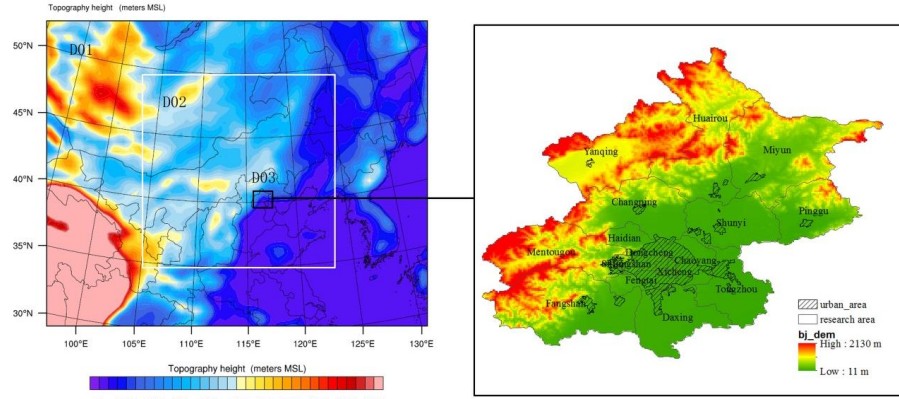

**Figure 1**: **Relative location and the geometry feature of the study area (the left one shows the three levels of nested domains adopted in most comparative experiments with D03 covering the whole Beijing area; the right one depicts the geometry features of the Beijing area).**



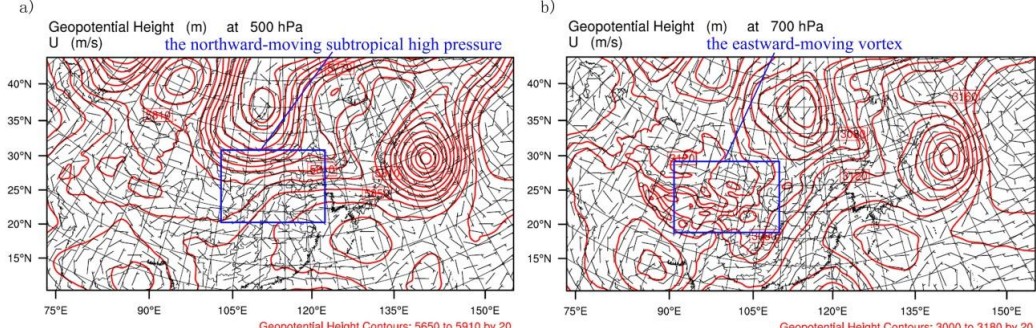

**Figure 2: Initial wind field and geopotential height field provided by the ERA-Interim reanalysis within D01 of C2 at 12 pm on 20 July 2012 (a) the fields at 500pha, (b) the fields at 700pha.**





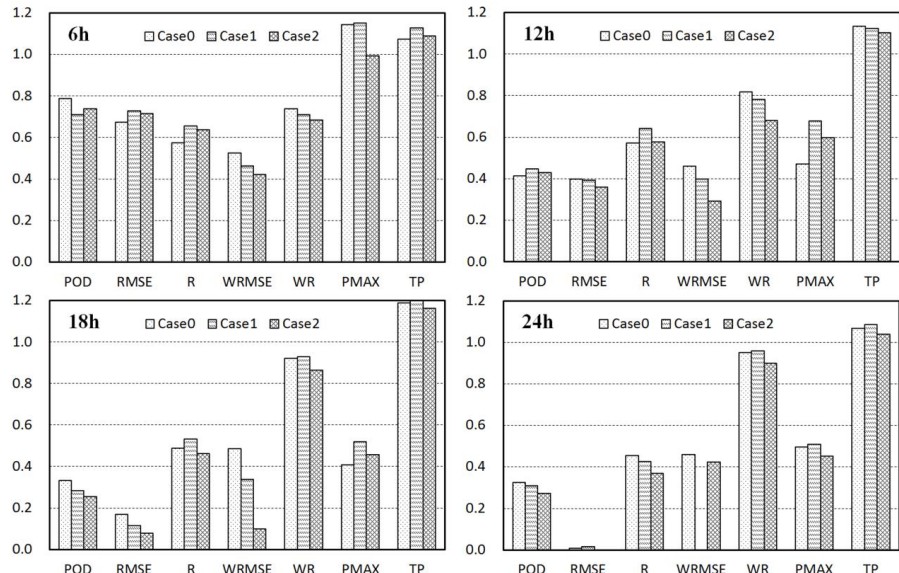

**Figure 3: Spatial values of the verification metrics for the WRF domain size experiments in S1 calculated within different temporal durations (Case 0 has the smallest domain size covering Northern-Central China; Case 1 has the intermediate domain size covering Northern China and a part of the Moangol Country; Case 2 has the largest domain size covering the northeastern hemisphere; the statistic time durations are 6 h, 12 h, 18 h, and 24 h, respectively, counting from 12 am 21 July 2012).**



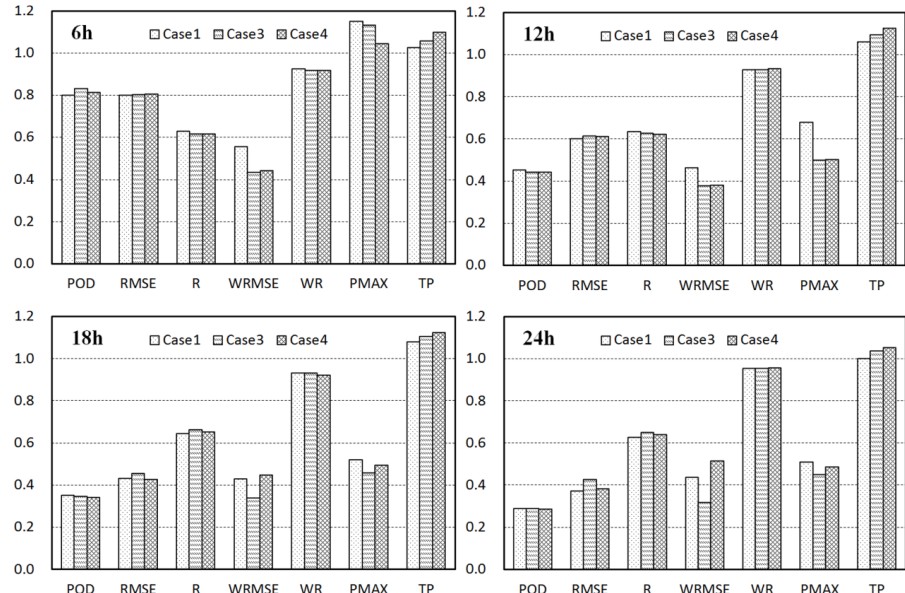

**Figure 4: As in Fig. 3, but for the experiments in S2 with different vertical resolution (Case 1 has 29 vertical levels equal to that of the ERA-Interim reanalysis, Case 3 and Case 4 has doubled and tripled vertical levels, respectively).**





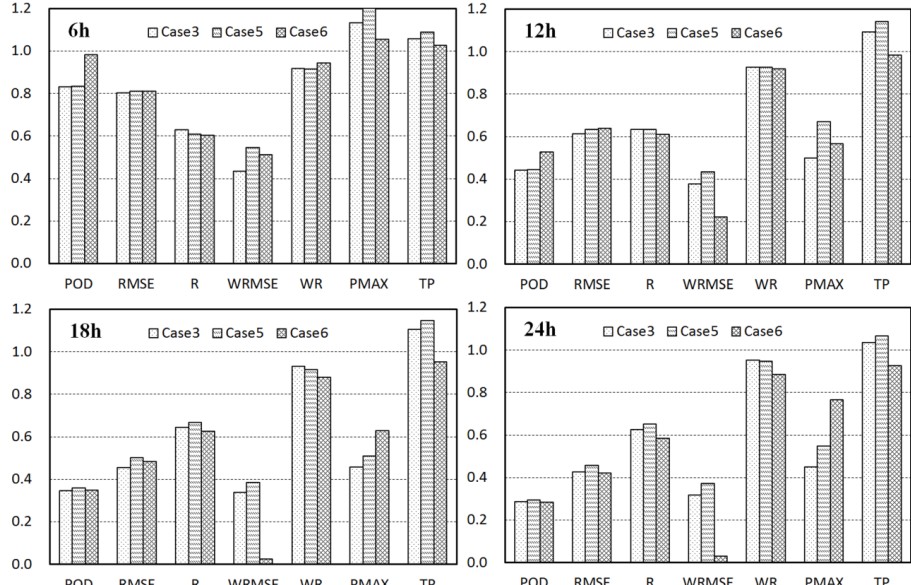

**Figure 5: As in Fig. 3, but for the experiments in S3 with different horizontal resolution (Case 3 has initial downscaling ratio of 1:3:3 with largest domain size of 40.5 km; Case 5 and Case 6 have the same largest domain size with 1:5:5 and 1:7:7 downscaling ratio, respectively).**



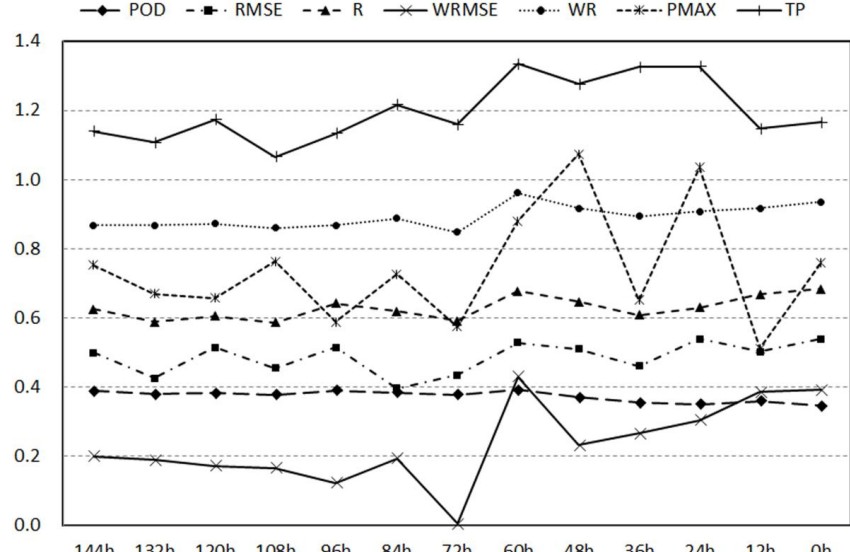

**Figure 6: Spatial values of the verification metrics for the WRF spin-up experiments in S4 calculated within 18 h time periods (Case 5 has the initial spin-up time of 12 h; Case 7 is designed with 0 h spin-up time; From Case 8 to Case 18, the spin-up time is increased by 12 h from 24 h).**



Table 1: Category of the experiments with different domain size, vertical resolution, horizontal resolution and spin-up time.

| Scenario | Experiment Number | Domain Size (grid numbers) | Vertical Levels | Horizontal Resolution (downscaling ratio) | Spin-up Time |
|---|---|---|---|---|---|
| Domain Size (S1) | Case 0 (C0) | D01 40×40 D02 72×72 | 29 | D01 40.5 km 1:3:3 | 12 h |
| | Case 1 (C1) | D01 80×64 D02 120×120 | As C0 | As C0 | As C0 |
| | Case 2 (C2) | D01 160×128 D02 240×192 | As C0 | As C0 | As C0 |
| Vertical Resolution (S2) | Optimal Case in S1 (OS1) | As OS1 | 29 | As C0 | As C0 |
| | Case 3 (C3) | As OS1 | 57 | As C0 | As C0 |
| | Case 4 (C4) | As OS1 | 85 | As C0 | As C0 |
| Horizontal Resolution (S3) | Optimal Case in S2 (OS2) | As OS1 | As OS2 | 1:3:3 | As C0 |
| | Case 5 (C5) | As OS1 | As OS2 | 1:5:5 | As C0 |
| | Case 6 (C6) | As OS1 | As OS2 | 1:7:7 | As C0 |
| Spin-up Time (S4) | Optimal Case in S3 (OS3) | As OS1 | As OS2 | As OS3 | 12 h |
| | Case 7 (C7) | As OS1 | As OS2 | As OS3 | 0 h |
| | Case8-Case 18 (C8-C18) | As OS1 | As OS2 | As OS3 | 24 h – 144 h per 12 h |





**Table 2: Correlations between the original value and the rescaled value of the verification metrics.**

| Original Value of the metrics | Rescaled Value of the metrics | Threshold Value |
|:---:|:---:|:---:|
| $POD$ | $POD/POD_{max}$ | + 0.115 max |
| $RMSE$ | $1 - RMSE/RMSE_{max}$ | + 41 max |
| $R$ | $R$ | N/A |
| $WRMSE$ | $1 - WRMSE/WRMSE_{max}$ | + 7.3 max |
| $WR$ | $WR$ | N/A |
| $RE_{PMAX}$ | $PMAX = RE_{PMAX}$ | N/A |
| $RE_{TP}$ | $TP = RE_{TP}$ | N/A |





**Table 3: Comparison of the metrics (18 h) between the initial experiment and the evaluated optimum experiments in each scenario.**

| Experiment Number | POD | RMSE | R | WRMSE | WR | PMAX | TP |
|---|---|---|---|---|---|---|---|
| Case 0 (C0) | 0.335 | 0.171 | 0.490 | 0.486 | 0.921 | 0.410 | 1.191 |
| Case 1 (C1) | 0.286 | 0.145 | 0.533 | 0.338 | 0.930 | 0.520 | 1.200 |
|  | 0.353 | 0.432 | 0.645 | 0.429 | 0.933 | 0.520 | 1.081 |
| Case 3 (C3) | 0.349 | 0.458 | 0.663 | 0.340 | 0.933 | 0.458 | 1.106 |
| Case 5 (C5) | 0.360 | 0.503 | 0.669 | 0.386 | 0.917 | 0.512 | 1.148 |
| Case 11 (C11) | 0.392 | 0.529 | 0.678 | 0.431 | 0.962 | 0.881 | 1.334 |