# Peer review of "Evaluation of the WRF model with different domain configurations"

_Hydrology and Earth System Sciences, 2017_

## Referee Comment (RC1) · Anonymous Referee #1 · 15 Sep 2017

**General Comments**

The authors present a study to investigate the impact of different domain sizes, vertical resolution, nesting ratios and spin-up time on a heavy precipitation event over Beijing. The simulations were forced by ERA-Interim reanalysis data available on 0.75° resolution in six hourly intervals. The different experiments were performed using three domains with a two-way nesting approach and the innermost domain centered on Beijing. Sub-daily precipitation of the second domain was verified against gridded precipitation observations from the China Meteorological Center. In addition, the precipitable water content of the WRF simulations was validated with ERA-Interim reanalysis data as a proxy for the maximum possible precipitation.

In a general sense, this type of experiments is of great relevance for flash flood forecasting and early warning systems. However, in the current experimental setup, I see several critical points preventing the traceability of the results.

1) The authors apply 2-way nesting from the outer to the inner domains. This means that precipitation patterns of the 3$^{rd}$ domain (which is not analyzed in your study) are reflected in the second domain. This actually means that you verify the precipitation from domain three mixed with terrain and land use data from domain two. Why did you verify domain two instead of domain three in your study? When looking at Fig. 1, Beijing has complex terrain which is not accurately represented at 13.5 km resolution.
Also with a 2-way nesting approach, you do not balance any kind of model physics with respect to the lateral boundary conditions. In a 2-way nesting approach, the fine grid resolution replaces the coarser scale resolution over the area of domain three.

2) The authors decided to use ERA-Interim reanalysis data to initialize their model simulations. As mentioned on page four, the resolution is 0.75°. I am not sure if such a coarse resolution is able to provide reasonable initial conditions, especially when focusing on sub-daily rainfall. Although you mention that a small domain may benefit from the lateral boundary conditions, I doubt that such a small outer domain of effectively 30*30 grid cells (due to boundaries of at least 5 cells in each direction) is sufficient here. This is also mentioned on page five in your manuscript. If you carefully checked the WRF webpage, you may have noticed that least 100*100 cells are recommended for every domain.

3) It is also not clear how the WRF model levels are distributed in your simulation. From table 1, I only see that you used 29 levels up to 50 hPa. If a constant grid spacing of 1 km is applied, your model simulations will fail because processes in the PBL are not at all resolved. If you are in the middle troposphere, this spacing can be sufficient. Also the WRF tutorial and website suggest a vertical grid spacing of **less** than 1km. If you look at the user guide (e.g. http://www2.mmm.ucar.edu/wrf/users/docs/user_guide_V3.8/users_guide_chap5.htm #examples) you will see that at least ~40 levels are recommended when the model top is

set to 50 hPa. As your outer domain gets enlarged towards the pole, how did you deal with the map factors?

Choosing an adaptive time step may save computation time but is not the best way for scientific experiments (see WRF webpage).

4) The rescaling of the error measures may lead to a misinterpretation. In case of POD, how did you chose the factor 0.115? This is not clear from the manuscript. Is the maximum RMSE used for each individual time step or is it calculated from an average over all the time steps? It is also hard to believe that the POD remains constant, independent whether you start one week or 12 hours before the event? Also, what is the precipitation threshold used to calculate POD? Is it 0.1mm? Usually, POD is applied for different thresholds.

How did you match both grids together? Did you use CDO, NCO, or NCL for this? It seems that you applied a $1/R^2$ approach to remap the CMC precipitation observations to the WRF grid. What is the radius of influence in this case? This can strongly determine the resulting field, especially in case of heavy and localized precipitation. Have the integrated water vapor fields been handled in the same way?

It would be very useful, if the authors provide horizontal plots of the precipitation patterns to substantiate the results. The applied scores do not necessarily tell if the precipitation is simulated spatially correctly.

In my opinion, a lot of important information is missing here and I also see deficiencies in the experimental setup. Therefore my recommendation is to reject the manuscript.

At the same time I encourage the authors to consider my suggestions and to resubmit a revised and updated version of the manuscript in the future. Before considering a resubmission, I strongly suggest that a native English speaker reads through the manuscript.

---

## Author Comment (AC1) · 9 Oct 2017

The revised paper has not been uploaded because we are waiting for the comments from other reviewers and the current version was sent to an English native speaker for checking now. But we want to respond to the comments received.

We would like to thank the referee for the helpful comments and thorough suggestions that will surely help us improve the manuscript. We will take all of them into consideration while revising the paper. Below, we provide our point-to-point response to the reviewer's comments and describe the modifications to address those issues.

Reviewer #1

"The authors present a study to investigate the impact of different domain sizes, vertical resolution, nesting ratios and spin-up time on a heavy precipitation event over Beijing. The simulations were forced by ERA-Interim reanalysis data available on 0.75 ° resolution in six hourly intervals. The different experiments were performed using three domains with a two-way nesting approach and the innermost domain centered on Beijing. Sub-daily precipitation of the second domain was verified against gridded precipitation observations from the China Meteorological Center. In addition, the precipitable water content of the WRF simulations was validated with ERA-Interim reanalysis data as a proxy for the maximum possible precipitation.

In a general sense, this type of experiments is of great relevance for flash flood forecasting and early warning systems. However, in the current experimental setup, I see several critical points preventing the traceability of the results."

**Response**: Thank you for pointing this out. We really appreciate your time and effort invested in the review. We have carefully checked all issues and added the missing information to increase the traceability of the results.

"1) The authors apply 2-way nesting from the outer to the inner domains. This means that precipitation patterns of the 3rd domain (which is not analyzed in your study) are reflected in the second domain. This actually means that you verify the precipitation from domain three mixed with terrain and land use data from domain two. Why did you verify domain two instead of domain three in your study? When looking at Fig. 1, Beijing has complex terrain which is not accurately represented at 13.5 km resolution."

**Response**: Thank you for pointing this out. Indeed, domain three covers the main convective process and should be the best choice for urban flooding study. We did attempt to verify domain three when we started this research. But then we noticed that the influence of some WRF model configurations, such as domain size and spin-up time, on simulating this heavy rainfall event was not well presented by this innermost domain.

[Figure]

**Figure 1. Spatial Distribution of 6 hours Accumulated Precipitation with different domain size within D02 during the heavy rainfall event from 12 pm, July 21, 2012 (a) precipitation of C0 with the smallest domain size in the first 6h (b) precipitation of C0 in the second 6h (c) precipitation ofC0 in the third 6h (d) precipitation of C1 with the medium domain size in the first 6h (e) precipitation of C1 in the second 6h (f) precipitation ofC1 in the third 6h (g) precipitation of C2 with the largest domain size in the first 6h (h) precipitation of C2 in the second 6h (i) precipitation of C2 in the third 6h.**

Taking the domain size for example (**Fig.1**), it can be seen that the differences shown in domain three are less obvious among these experiments, while the spatial pattern significantly differs when looking at domain two. Besides, the spatial resolution of the CMC ground observation dataset, the one that is publicly available with the highest spatial resolution of 0.1 degrees, approximates to the resolution of domain two. Therefore, we compromised to extend the analyzed range to domain two for comparison in the submitted version to illustrate the possible influence on the whole pattern of the rain-belt during this rain storm. We are sorry for not making it clear in our previous manuscript. The reason for choosing domain

two for verification will be added to the revised version. Besides, the representative results of domain three were provided in the discussion section to illustrate this issue.

"Also with a 2-way nesting approach, you do not balance any kind of model physics with respect to the lateral boundary conditions. In a 2-way nesting approach, the fine grid resolution replaces the coarser scale resolution over the area of domain three."

**Response**: Thank you for pointing it out. We are sorry for not carefully checking this statement. Indeed, in a two-way nesting approach, the parent domain has its independent run, but it serves the child domain's boundary condition at each time step. And then after the child domain's dynamic modeling run, the child domain result (including the rainfall field) is mapped onto the target parent domain grids. We will remove this confusing sentence in the revised manuscript.

"2) The authors decided to use ERA-Interim reanalysis data to initialize their modelsimulations. As mentioned on page four, the resolution is 0.75°. I am not sure if such acoarse resolution is able to provide reasonable initial conditions, especially when focusing on sub-daily rainfall."

**Response**: Thank you for raising this question. We agree that this dataset maynot be the best choice if other sources of input fields with higher resolution were available. However, the ERA-Interim reanalysis dataset has been widely used in downscaling studies with acceptable results, and it is the best source of the reanalysis data accessible in the study area.

"Although you mention that a small domain may benefitfrom the lateral boundary conditions, I doubt that such a small outer domain ofeffectively 30*30 ($40\times40$) grid cells (due to boundaries of at least 5 cells in each direction) issufficient here. This is also mentioned on page five in your manuscript. If you carefullychecked the WRF webpage, you may have noticed that at least 100*100 cells arerecommended for every domain."

**Response**: Thank you for pointing it out. In WRF-ARW, at least five cells along the boundaries of each domain are indeed required to mitigate sharp gradients (i.e., short wavelength features) that may exist along the lateral boundaries where the specified lateral boundary conditions differ from their values on the interior of the limited-area domain. As you pointed out, we carefully rechecked our settings of all the experiments to make sure that we followed this recommendation. However, in our smallest domain size experiment (C0), eight cells are set between the outermost grid and the interim grid, 11 cells are set between the interim grid and the innermost grid, and 18 cells are set in the innermost grid in both x- and y-direction. We believe that these distances are sufficient for relaxation.

As to the domain size of at least $100\times100$, this should be related to the weather system in the study area. As long as the domain size covers the key weather features, it should be fine with a smaller domain than $100\times100$. Examples can be seen in Figure 3c of Bukovsky and Karoly (2009) where $45\times70$ cells were adopted in the outer domain covering the U.S. and

in Figure 1a of Dasari et al. (2014) where 98×55 cells were used in the outer domain covering most of Europe. In our initial case (C0), the main feature of the weather systems that led to this storm was included within the outermost domain. Besides, to estimate the potential influence of lateral boundary conditions to the rainfall outputs, two other cases with larger domain size were designed for comparison.

Bukovsky, M. S., and Karoly, D. J.: Precipitation simulations using WRF as a nested regional climate model. *J. Appl. Meteorol. Climatol.*, **48(10)**, 2152-2159, doi:10.1175/2009JAMC2186.1, 2009.

Dasari, H. P., Salgado, R., Perdigao, J., and Challa, V. S.: A regional climate simulation study using WRF-ARW model over Europe and evaluation for extreme temperature weather events. *Intl. J. Atmospheric Sci.*, **9**, 2101-2122, doi:10.1155/2014/704079, 2014.

"3) It is also not clear how the WRF model levels are distributed in your simulation. From table 1, I only see that you used 29 levels up to 50 hPa. If a constant grid spacing of 1 km is applied, your model simulations will fail because processes in the PBL are not at all resolved. If you are in the middle troposphere, this spacing can be sufficient. Also the WRF tutorial and website suggest a vertical grid spacing of less than 1km. If you look at the user guide (e.g. http://www2.mmm.ucar.edu/wrf/users/docs/user guide V3.8/users guide chap5.htm#examples) you will see that at least ~40 levels are recommended when the model top isset to 50 hPa."

**Response**: Thank you for pointing it out. Indeed, vertical grid spacing of less than 1km was suggested, especially in the lower and middle troposphere layer where convective process mainly happens. In this study, nonuniform grid spacing was adopted, and the eta value for WRF was calculated by using the vertical eta levels provided by the ERA-Interim. The detailed configuration can be seen in **Table 1**.

**Table 1. Vertical levels set in the initial experiment and the corresponding height in Beijing, China.**

| Eta Level | 1 | 2 | 3 | 4 | 5 | 6 | 7 | 8 | 9 | 10 | 11 | 12 | 13 | 14 | 15 |
|---|---|---|---|---|---|---|---|---|---|---|---|---|---|---|---|
| Pressure (hpa) | 1000 | 975 | 950 | 925 | 900 | 875 | 850 | 825 | **800** | 775 | 750 | 700 | 650 | 600 | 550 |
| Eta value (0-1) | 1 | 0.973 | 0.947 | 0.921 | 0.894 | 0.868 | 0.842 | 0.815 | 0.789 | 0.763 | 0.736 | 0.684 | 0.631 | 0.578 | 0.526 |
| Height (km) | 0 | | | | 0.988 | | 1.457 | | 1.949 | | | 3.012 | | 4.206 | |

| Eta Level | 16 | 17 | 18 | 19 | 20 | 21 | 22 | 23 | 24 | 25 | 26 | 27 | 28 | 29 |
|---|---|---|---|---|---|---|---|---|---|---|---|---|---|---|
| Pressure (hpa) | 500 | 450 | 400 | 350 | 300 | **250** | 225 | 200 | 175 | 150 | 125 | 100 | 70 | 50 |
| Eta Value (0-1) | | | | | | | | | | | | | | |
| Height (km) | 5.574 | | 7.185 | | 9.164 | 10.363 | | 11.784 | | | | 16.18 | 18.442 | 20.576 |

Here, 1000pha-800pha corresponds to the lower troposphere; 800pha-250pha corresponds to the middle and upper troposphere. The approximation of the height was also provided by referring the summer value provided by a ground study in Beijing. By comparing the values, we can see that the setting of vertical levels meets the requirements. We are sorry for not making it clear in the manuscript. The detailed illustration will be added to the revised manuscript.

"As your outer domain gets enlarged towards the pole, how did you deal with the map factors?"

**Response**: Thank you for pointing it out. In this study, our main focus is on the simulation in domain two and domain three which are located in the medium latitude of the northern hemisphere, so the Lambert conformal projection was selected and centered at the same latitude (42.25°N) and longitude (114.0°E).

[Figure]

**Figure 2. The relative location of the nested domains adopted in C2 that with the largest outer domain of all the experiments.**

**Figure 2** shows the domain configuration with the largest outer domain. In this figure, it can be seen that the boundary of the outer domain is still far away from the pole and centered at the medium latitude area. The above-mentioned info will be added to the revised manuscript.

"Choosing an adaptive time step may save computation time but is not the best way forscientific experiments (see WRF webpage)."

**Response**: Thank you for raising this question. As you mentioned, choosing an adaptive time step may not be the best way for scientific experiments. We are sorry for not carefully checking this statement and giving the impression that all the experiments in this study adopt the adaptive time step. In fact, C4 with the finest vertical resolution has used this setting, as well as C6 with the finest horizontal resolution. When running these two cases, the recommended minimum time step was set (about $3 \times DX$ seconds for the outermost domain), but instability was encountered, and the model ran much slower than we expected and stopped before the end time. To deal with this problem, an adaptive time step is adopted, where the maximum time step was up to $6 \times DX$ with CFL value set to 1.2.

"4) The rescaling of the error measures may lead to a misinterpretation. In case of POD, howdid you choose the factor 0.115? This is not clear from the manuscript."

**Response**: Thank you for pointing it out. It is our negligence to forget adding the related reference to illustrate how the scale for each verification parameter was selected. Here, we add two references in which the same method was adopted to

allow a convenient andmultidimensional assessment of precipitationsimulation quality for various WRF configurations. Please see Table 2 in Sikder and Hossain (2016).

In the case of the POD, the factor of 0.115 is determined by the largest POD value calculated from all the experiments. Then all the POD values were divided by this factor to ensure they fell within the range of 0-1. The detailed illustration and reference will be added to the revised paper.

Sikder, S., and Hossain, F.: Assessment of the weather research and forecasting model generalized parameterization schemes for the advancement of precipitation forecasting in monsoon-driven river basins, *J. Adv. Model. Earth Syst.*, **8**, 1210–1228, doi:10.1002/2016MS000678, 2016.

"Is the maximum RMSE used for each individual time step or is it calculated from an average over all thetime steps?"

**Response**: Thank you for raising this question. The RMSE was calculated through the same process as POD. The procedure has been illustrated by Tian et al. (2016) in comparing the simulations to the observations from the ground meteorological stations. As the temporal duration was shorter and the spatial calculation was at grid scale, we merely mentioned the difference when we adopted the method. In this study, all the metrics were firstly computed between the observations and simulations of the same grid at each time step and then averaged within different time durations (6h, 12h, 18h, 24h) for the final analysis.

Tian, J. Y., Liu, J., Li, C. Z., and Yu, F. L.: Numerical rainfall simulation with different spatial and temporal evenness by using WRF multi-physics ensembles. *Nat. Hazards Earth Syst. Sci. Discuss.*, **17**, 563-579, doi:10.5194/nhess-17-563-2017, 2017.

"It is also hard to believe that the POD remains constant, independentwhether you start one week or 12 hours before the event?"

**Response**: Thank you for raising this question. In our study, as the POD was averaged among the large domain with at least $72 \times 72$ grids. The differences presented could be less obvious than expected. Besides, when shown in the same graph with the other parameters, the differences were much less obvious than others. But it doesn't mean it is constant.

"Also, what is the precipitation threshold used to calculate POD? Is it 0.1mm? Usually, POD is applied for different thresholds."

**Response**: Thank you for raising this idea. In this study, 0.1 mm is used to calculate POD. POD with different thresholds may be useful to investigate the accurate hit of the heavy rainfall area further. Considering that RMSE and R could also reflect this feature, we only choose POD with 0.1mm as one of the verification parameters in this study.

"How did you match both grids together? Did you use CDO, NCO, or NCL for this? It seems that you applied a $1/R^2$ approach to remap the CMC precipitation observations to the WRF grid. What is the radius of influence in this case? This can strongly determine the resulting field, especially in case of heavy and localized precipitation."

**Response**: Thank you for pointing it out. We are sorry to miss the detailed illustration of the interpolation process. Here, for the rainfall values, the bi-linear interpolation method was adopted to remap the CMC precipitation observations (about 10 km×10 km) to the WRF grid (including 13.5 km×13.5km, 8.1 km×8.1 km and one with 5.785km×5.785km). In this method, four nearest points for each WRF grid were searched to accomplish the bi-linear interpolation process. And the bi-linear interpolation method is now the default option used by WRF-ARW to interpolate the initial meteorological fields.

"Have the integrated water vapor fields been handled in the same way?"

**Response**: Thank you for raising this question. The initial water vapor field was extracted from WPS outputs by adding TCWV (Total column water vapor) in the Vtable files. This means that it has the same location as the field calculated from the WRF outputs. Therefore, there is no need to remap one field to another.

"It would be very useful, if the authors provide horizontal plots of the precipitation patterns to substantiate the results. The applied scores do not necessarily tell if the precipitation is simulated spatially correctly."

**Response**: Thank you for your suggestion. The plots of the spatial precipitation patterns indeed would be helpful to substantiate the results, such as the examples of domain size experiments. Some representative plots will be added to the revised manuscriptto make the results with more clarity.

"In my opinion, a lot of important information is missing here and I also see deficiencies in the experimental setup. I strongly suggest that a native English speaker reads through the manuscript."

**Response**: We hope our replies have addressed your concerns. The revised manuscript will be thoroughly proof-read by a native English speaker.

---

## Referee Comment (RC2) · Anonymous Referee #2 · 15 Oct 2017

In this contribution, the authors evaluate the performance of the WRF model in different configurations for a single heavy rainfall event centered over Beijing, China. The evaluation differs from other studies in that field in the sense that no physics parameterization evaluation is attempted. Instead, the model setup (domain configuration, number of vertical levels = vertical resolution, nesting ratio = horizontal resolution, forecasting lead time) are explored. In the order of the above, the best configuration is chosen in each step to perform several experiments in the next step. Several verification measures are employed for precipitation and precipitable water.

The design of these experiments are convincing despite a few weak points listed below.

[Figure]

The use of English language, however, needs improvement. Grammatical mistakes and strange wordings render some parts of the text unclear. I did not make any attempt to correct this, but highlight a few common issues below. With improvements to the language and several changes to the contents, the contribution may be suitable for publication.

General remarks

- Dependence of parameters varied: Although discussed in the introduction, the dependence of optimal lead forecasting time on domain extent (and vice versa) is not considered in the study. Instead, based on a standard lead time of 12h, the "best" domain configuration is derived as C1, based on which an optimal lead time of 60h is inferred later on as C11. In my understanding, these two lead times should match if one really found the "best" combination of these two parameters

- The authors choose an adaptive time step to conduct their modeling experiments. This introduces another free parameter, since the actual time step adopted in each simulation may vary and as such influence the results

- The performance evaluation of the WRF model is performed over the intermediate domain D02 and not over the highest-resolution domain D01. It is argued that the two-way nesting approach does inform D02 about the results in the innermost domain D03, but interpolation to the coarser D02 grid and the (possible) difference in setup of the model physics (see next point) may influence the conclusions drawn. I would like to encourage the authors to conduct simulations without D03 for their best setup at least

- Model physics: It is unclear to me whether the GD cumulus parameterization is also employed in D03 at convection-permitting resolution (<5-10km)

- The forcing data is obtained from ERA-Interim on pressure levels (28 levels). This is not ideal, in particular since the authors are trying to assess the added value of a higher vertical resolution and since ERA-Interim is also available on 38 model levels.

I would like to encourage the authors to repeat experiments with their optimal setup C11 using ERA-Interim model-level data and varying the vertical resolution as in S2, for example.

- Several of the abbreviations in the text or the figure captions are not introduced before they are used (or not at all), please check and correct

- The statistical measures used here have different directions of "good" (i.e. RMSE is good if low, R is good if high). This is not mentioned anywhere in the text and in the figures, which makes the interpretation confusing, also because the statistics are rescaled. I would encourage to state explicitly what value implies a better model performance for which statistical measure, and possibly encode this (in color or differently) in the figures and the tables

- Beneath dynamical downscaling explored here, also statistical downscaling methods and new global modeling approaches on irregular grids (e.g. MPAS) have been used and show promising results to forecast extreme precipitation events. This should be discussed briefly in the introduction or discussion section

Specific remarks

- Language-specific: - the word "occurred" is often missing a leading "that" or "which" - the expression "demonstrated true" seems odd to me - several times in the text, "grid" is used whereas "grid point" should be - singular and plural, as well as the use of articles need to be checked carefully

- Page 2, lines 24-26: WRF being used at resolutions >10km only is not true. Many leading operational NWP centers are employing WRF at convection-resolving resolution (NCEP: HRRR, 3km over CONUS; MeteoGroup: 3km over Europe; New Zealand MetService: <4km over New Zealand) operationally

- Page 4, line 10: "coaster-scale" -> "coarser-scale"

- Page 4, line 21: "Earth-system system"

- Page 4, line 27: Isn't ERA-Interim available from 1979 (not 1989)?

- Page 4, line 29: "WRFV3.7.2" or "WRFV3.7.1" ?

- Page 5, line 6: the correct reference should be Skamarock et al. (2008)

- Page 5, line 14: "They two together" -> "The two together"

- Page 7, line 31: RRTMG schemes (not RRTM)? Or is it "RRTM" for lw, and "Dhudia" for sw?

- Page 13, line 11: "less sensititvity" -> "less sensitive"

- Page 14, lines 2-15: the discussion is confusing for the reader as he/she is expected to translate nesting ratios into effective horizontal resolution. In this paragraph, as well as in Table 1, the effective horizontal resolution should be stated explicitly, alongside with the nesting ratios

- Page 14, line 31 to page 15, line 1: the authors state that the positive bias in pre-cipw depends on the initialization time, with largest biases for initialization times with highest amounts of precipw. This, in my opinion, is an important finding and should be highlighted and possibly discussed further

- Page 17, lines 8-11: The authors briefly discuss the dependence on the quality of the forcing data. This highlights the importance to conduct additional experiments with ERA-Interim model level data as described above, and at least discuss (if not evaluate) potential effects when using ECMWF high-res forecasts on 137 model levels and approx. 9km horizontal resolution

- Example for a necessary rewording: caption of figure 1: "Location and topography of the study area. Left panel: three levels of nested domains adopted in most experi-ments, with D03 covering the Beijing area; right panel: zoom-in on the topography of the Beijing area"

---

## Author Comment (AC2) · 7 Nov 2017

The revised paper has not been uploaded because we are waiting for the comments from other reviewers and the current manuscript was under checking by a native English speaker presently. But we want to respond to the comments received.

We would like to thank the referee for the helpful comments. Our point-to-point response to the reviewer's comments is described as follows.

Reviewer #2

"In this contribution, the authors evaluate the performance of the WRF model in different configurations for a single heavy rainfall event centered over Beijing, China. The evaluation differs from other studies in that field in the sense that no physics parameterization evaluation is attempted. Instead, the model setup (domain configuration, number of vertical levels = vertical resolution, nesting ratio = horizontal resolution, forecasting lead time) are explored. In the order of the above, the best configuration is chosen in each step to perform several experiments in the next step. Several verification measures are employed for precipitation and precipitable water (PW).

The design of these experiments is convincing despite a few weak points listed below. The use of English language, however, needs improvement. Grammatical mistakes and strange wordings render some parts of the text unclear. I did not make any attempt to correct this but highlight a few common issues below. With improvements to the language and several changes to the contents, the contribution may be suitable for publication."

**Response**: Thanks very much for the encouraging feedback.

**General remarks**

- Dependence of parameters varied: Although discussed in the introduction, the dependence of optimal lead forecasting time on domain extent (and vice versa) is not considered in the study. Instead, based on a standard lead time of 12h, the "best" domain configuration is derived as C1, based on which an optimal lead time of 60h is inferred later on as C11. In my understanding, these two lead times should match if one really found the "best" combination of these two parameters."

**Response**: We agree that there is a dependence relationship between domain extent and forecasting lead time and this is particularly true in the limited-area modeling cases where data assimilation is not conducted. For instance, a model run with a larger domain size may need longer lead time to spin-up physical processes of interest, such as clouds, precipitation, local ageostrophic circulations, and lateral boundary conditions. Besides, the choice of lead time and domain size at the same time determine the moment at which the initial and lateral boundary conditions are derived and the range of the corresponding synoptic features and water vapor conditions involved at that moment. This, however, means that for a given domain size, the results may also be affected by the degree of similarity between the forcing data and the real conditions at the initialization time. In addition, the choice of updating lateral boundary conditions at a given interval could

affect the time needed to spin-up the physical processes. It is noteworthy that although C1 is detected with the best performance, C2 with nearly doubled domain extent of C1 only differs obviously in the PW fields but with less diversity regarding distribution characteristics of the rain belt. Combining all the aforementioned factors, we believe that it makes sense that C11 (with 60h lead time) is evaluated with better performance when compared with C5 (with 12h lead time).

- The authors choose an adaptive time step to conduct their modeling experiments. This introduces another free parameter, since the actual time step adopted in each simulation may vary and as such influence the results."

**Response**: As you mentioned, choosing an adaptive time step may introduce another free parameter and as such influence the results. We are sorry for not carefully checking the statement and giving the impression that all experiments adopted the adaptive time step. In fact, only C4 with the finest vertical resolution, and C6 with the finest horizontal resolution have used this setting. When running these two cases, the recommended minimum time step was set (about $3 \times DX$ seconds for the outermost domain), but instability was encountered, and the model ran much slower than we expected and stopped before the end time. To deal with this problem, an adaptive time step was adopted, where the maximum time step was up to $6 \times DX$ with CFL value set to 1.2. We will edit this confusing statement and discuss it in the revised manuscript.

- The performance evaluation of the WRF model is performed over the intermediate domain D02 and not over the highest-resolution domain D03. It is argued that the two-way nesting approach does inform D02 about the results in the innermost domain D03, but interpolation to the coarser D02 grid and the (possible) difference in setup of the model physics (see next point) may influence the conclusions drawn. I would like to encourage the authors to conduct simulations without D03 for their best setup at least.

**Response**: Thank you for raising this question. Indeed, domain D03 covers the main convective processes and should be the best choice for evaluation. However, when we conducted the evaluation over domain D03, we noticed that the influence of some WRF model configurations, such as domain size and spin-up time, on simulating this heavy rainfall event was not well presented by this innermost domain. Besides, the spatial resolution of the CMC ground observation dataset, the one that is publicly available with the highest spatial resolution of 0.1 degrees, approximates to the resolution of domain D02. Therefore, we compromised to extend the analyzed range to domain D02 for comparison in the submitted version to illustrate the possible influence on the whole pattern of the rain-belt during this rain storm.

Meanwhile, we agree that the two-way nesting may alter the result in D02 (See **Fig. 1**) and it would be useful to explore the one-way nesting for comparison. We are sorry for not making this clear in our previous manuscript. The reason for choosing domain D02 for verification will be added to the revised manuscript.

[Figure]

**Figure 1. Spatial distribution of six hour accumulated precipitation for C11 (the experiment with the best setup) with different nesting schemes within D02 during the heavy rainfall event from 12 pm, July 21, 2012 (a) precipitation of C11 with two-way nesting in the first 6h; (b) precipitation of C11 with two-way nesting in the second 6h; (c) precipitation of C11 with two-way nesting in the third 6h; (d) precipitation of C11 with one-way nesting in the first 6h; (e) precipitation of C11 with one-way nesting in the second 6h, and (f) precipitation of C11 with one-way nesting in the third 6h.**

- Model physics: It is unclear to me whether the GD cumulus parameterization is also employed in D03 at convection-permitting resolution (<5-10km).

**Response**: Thank you for raising this question. In this study, GD cumulus parameterization was turned on for each domain, including D03 at the convection-permitting resolution between 1km and 5km, to represent the effects of sub-grid scale convective processes which were also detected in the rainfall processes of this heavy rainfall event.

- The forcing data is obtained from ERA-Interim on pressure levels (28 (29) levels). This is not ideal, in particular since the authors are trying to assess the added value of a higher vertical resolution and since ERA-Interim is also available on 38 model levels. I would like to encourage the authors to repeat experiments with their optimal setup C11 using ERA-Interim model-level data and varying the vertical resolution as in S2, for example.

**Response**: Thank you for your suggestion. We agree that the experiment with slightly higher vertical resolutions could get better performance than the one with lower vertical resolution. This has also been verified in this study where the experiment with 57 vertical levels shows better performance than the one with 29 vertical levels. In this study, the ERA-Interim 29 pressure level data was selected as the initial forcing for two reasons. First, it meets the requirements of less than 1 km distance between each vertical level in the free troposphere where convective processes mainly happen (See **Table 1**). Second, the NWP models used by Chinese Meteorological Center are mainly operated with 31 vertical levels for regional forecasting (See Table 11-2.1 in WMO, 2013).

**Table 1. The vertical levels set in the initial experiment and the corresponding height in Beijing, China.**

| Eta Level | 1 | 2 | 3 | 4 | 5 | 6 | 7 | 8 | 9 | 10 | 11 | 12 | 13 | 14 | 15 |
|---|---|---|---|---|---|---|---|---|---|---|---|---|---|---|---|
| Pressure (hpa) | 1000 | 975 | 950 | 925 | 900 | 875 | 850 | 825 | **800** | 775 | 750 | 700 | 650 | 600 | 550 |
| Eta value (0-1) | 1 | 0.973 | 0.947 | 0.921 | 0.894 | 0.868 | 0.842 | 0.815 | 0.789 | 0.763 | 0.736 | 0.684 | 0.631 | 0.578 | 0.526 |
| Height (km) | 0 | | | | | 0.988 | | 1.457 | | 1.949 | | | 3.012 | | 4.206 |

| Eta Level | 16 | 17 | 18 | 19 | 20 | 21 | 22 | 23 | 24 | 25 | 26 | 27 | 28 | 29 |
|---|---|---|---|---|---|---|---|---|---|---|---|---|---|---|
| Pressure (hpa) | 500 | 450 | 400 | 350 | 300 | **250** | 225 | 200 | 175 | 150 | 125 | 100 | 70 | 50 |
| Eta Value (0-1) | | | | | | | | | | | | | | |
| Height (km) | 5.574 | | 7.185 | | 9.164 | 10.363 | | 11.784 | | | | 16.18 | 18.442 | 20.576 |

However, although the ERA-Interim 29 pressure level data meets the requirement when simulating the heavy rainfall event in Beijing, it doesn't mean that it satisfies the condition in other regions. To illustrate this issue, one experiment forced with ERA-Interim 38 pressure level data will be added in S2 for comparison. Besides, the possible effect of the selection of initial forcing data on the forecasts will be discussed in the revised manuscript.

WMO: Anticipated advances in numerical weather prediction, and the growing technology gap in weather forecast. 2013.[Available online at https://www.wmo.int/pages/prog/www/swfdp/Meetings/documents/Advances_NWP.pdf]

- Several of the abbreviations in the text or the figure captions are not introduced before they are used (or not at all), please check and correct

**Response**: Thank you for pointing it out. We will check all the abbreviations in the text and figure captions and make sure their full names will be added before the abbreviations are used.

- The statistical measures used here have different directions of "good". (i.e. RMSE is good if low, R is good if high). This is not mentioned anywhere in the text and in the figures, which makes the interpretation confusing, also because the statistics are rescaled. I would encourage to state explicitly what value implies a better model performance for which statistical measure, and possibly encode this (in color or differently) in the figures and the tables

**Response**: Thank you for raising this question. To make it clear, the verification metrics will be assigned with a new set of symbols after the statistics are rescaled (see **Table 2**). The explanation of the performance measures will be added in the revised manuscript as suggested.

**Table 2. Correlations between original and rescaled verification metrics.**

| Original Metrics | Representative Meaning | Rescaled Metrics | Threshold Value |
|---|---|---|---|
| $POD$ | Probability of Detection | $POD' = POD/POD_{max}$ | + 0.115 max |
| $RMSE$ | Root Mean Square Error | $RMSE' = 1 - RMSE/RMSE_{max}$ | + 41 max |
| $R$ | Pearson Correlation Coefficients | $R' = R$ | N/A |
| $WRMSE$ | RMSE of the Precipitable Water | $WRMSE' = 1 - WRMSE/WRMSE_{max}$ | + 7.3 max |
| $WR$ | R of the Precipitable Water | $WR' = WR$ | N/A |
| $RE_{PMAX}$ | Relative Error of the Maximum Precipitation | $PMAX' = RE_{PMAX}$ | N/A |
| $RE_{TP}$ | Relative Error of the Total Precipitation | $TP' = RE_{TP}$ | N/A |

- Beneath dynamical downscaling explored here, also statistical downscaling methods and new global modeling approaches on irregular grids (e.g. MPAS) have been used and show promising results to forecast extreme precipitation events. This should be discussed briefly in the introduction or discussion section”

**Response**: Thank you for your suggestion. We agree that some statistical downscaling methods, along with data assimilation methods, could provide more reliable forecasts of extreme precipitation events, especially for short-term forecasting with 6h-24h lead time. The related content will be added in the discussion section as suggested.

**Specific remarks**

- Page 2, lines 24-26: WRF being used at resolutions >10km only is not true. Many leading operational NWP centers are employing WRF at convection-resolving resolution (NCEP: HRRR, 3km over CONUS; Meteo Group: 3km over Europe; New Zealand Met Service: <4km over New Zealand) operationally.

**Response**: Thank you for the clarification. We will rephrase this sentence in the manuscript to avoid the confusion.

- Page 4, line 27: Isn't ERA-Interim available from 1979 (not 1989)?

**Response**: Thank you for pointing this out. Indeed, ERA-Interim is available from 1979 (originally, ERA-Interim ran from 1989, but the 10-year extension for 1979-1988 was added in 2011). We will amend this statement.

- Page 7, line 31: RRTMG schemes (not RRTM)? Or is it "RRTM" for LW and "Dhudia" for SW?

**Response**: Thank you for pointing this out. The radiation schemes adopted in our study were the RRTMG schemes. We will edit this sentence to: “The radiation processes were represented by the RRTMG short-wave radiation and the RRTMG long-wave radiation schemes (Iacono et al., 2008), respectively.”

- Page 14, lines 2-15: the discussion is confusing for the reader as he/she is expected to translate nesting ratios into effective horizontal resolution. In this paragraph, as well as in Table 1, the effective horizontal resolution should be stated explicitly, alongside with the nesting ratios"

**Response**: Thank you for the suggestion. We will follow the advice, as noted above.

- Page 14, line 31 to page 15, line 1: the authors state that the positive bias in precipw (PMAX) depends on the initialization time, with largest biases for initialization times with highest amounts of precipw. This, in my opinion, is an important finding and should be highlighted and possibly discussed further.

**Response**: Thanks for your suggestion. We agree that this is an interesting finding that should be highlighted and further discussed as suggested.

- Page 17, lines 8-11: The authors briefly discuss the dependence on the quality of the forcing data. This highlights the importance to conduct additional experiments with ERA-Interim model level data as described above, and at least discuss (if not evaluate) potential effects when using ECMWF high-res forecasts on 137 model levels and approx. 9km horizontal resolution.

**Response**: Thank you for pointing this out. As mentioned above, one experiment forced with ERA-Interim 38 model level data will be added in Scenario 2 for comparison. Besides, the possible effect of the quality of the forcing data on the forecasts will be further discussed in the revised manuscript.

**Typographical corrections**

- Page 4, line 10: "coaster-scale" -> "coarser-scale"

- Page 4, line 21: "Earth-system system"

- Page 4, line 29: "WRFV3.7.2" or "WRFV3.7.1"

- Page 5, line 14: "They two together" -> "The two together"

- Page 5, line 6: the correct reference should be Skamarock et al. (2008).

- Page 13, line 11: "less sensititvity" -> "less sensitive"

**Response**: Thanks for reading our manuscript so meticulously – these will be corrected.

**Grammar corrections**

- Language-specific: - the word "occurred" is often missing a leading "that" or "which" -the expression "demonstrated true" seems odd to me - several times in the text, "grid" is used whereas "grid point" should be - singular and plural, as well as the use of articles need to be checked carefully.

- Example for a necessary rewording: caption of figure 1: "Location and topography of the study area. Left panel: three levels of nested domains adopted in most experiments, with D03 covering the Beijing area; right panel: zoom-in on the topography of the Beijing area"

**Response:** Thank you for pointing this out. We will carefully check through the manuscript to correct the grammatical mistakes with more precise descriptions.

We hope our replies have addressed your concerns, and the revised manuscript will be thoroughly proof-read by a native English speaker.

---

## Editor Comment (EC1) · U. Ehret (Editor) · 29 Nov 2017

**Editor comment for manuscript hess-2017-363**

**Evaluation of the WRF model with different domain configurations and spin-up time in reproducing a sub-daily extreme rainfall event in Beijing, China**

**by Qi Chu et al.**

Dear Authors,

I have read the referees' comments and your replies. While for most points, your replies are satisfactory, I need some more information from you about a major point both referees raised: The choice of modeling and evaluation domains (comment 2 of referee #1, comment 4 of referee #2). In the following, I will explain my current understanding of what you were doing, and my related conclusions. Before I take my decision about how to proceed, I would like to verify that I understood things correctly:

- You model an extreme rainfall event using WRF set up on 3 nested domains (D01, D02, D03) with D01 being the largest domain in coarsest resolution, D03 the smallest, which covers the area of interest, Bejing region.
- When running the models, D01 is forced by ERA-Interim reanalysis, D02 by D01, D03 by D02
- For analysis/evaluation, you map rainfall from D03 back into the results and onto the grid of the D02 domain.
- Analysis and evaluation is the done with this hybrid data set D02+partlyD03, on the D02 domain, against ground rainfall observations and ERA-interim reanalysis.
- The reasons for doing so is that a) the reference truth is available only in a resolution comparable to the D02 resolution, and b) differences among the models are less obvious in the D03 domain than in the D02 domain

So if this is correct, I have two main concerns:

- If your goal is to evaluate different WRF setups with respect to regional (here: D03 or Bejing-scale) heavy precipitation, then a) the evaluation should be done exactly on this scale and b) they should be compared to reference data with adequate resolution for that scale.
- If your goal is to evaluate different WRF setups with respect to larger-scale (here: D02-scale) precipitation patterns, then the D03 run on its small scale is unnecessary. You could then run WRF on the D02 domain with different configurations (the D02 and the D03 settings with respect to parameterization, grid size etc.) and compare these.

So if my concerns are based on correct understanding, you will either have to

- gather better reference data and repeat the evaluation on the D03 scale, or
- do new model runs on the D02 scale and repeat evaluation on the D02 scale

For either way, I feel that more than the time normally spent on major revisions is necessary. As I still think your study is relevant and interesting, I see three possible ways from here: Either you are confident to finish the required work within, 2-3 months, then the decision can be major revisions. If not, you could withdraw and resubmit when you are ready, or I reject the study with the invitation for resubmission. Please let me know what you prefer.

Yours sincerely,
Uwe Ehret

---

## Author Comment (AC3) · 20 Dec 2017

Dear Dr. Ehret,

Thank you very much for your positive assessment and constructive suggestions to our study. After carefully reading your comments, we do agree that an in-depth analysis should be made on the rationality of our choice of the modeling and evaluation domains. Therefore, we have repeated the analysis on the D03 scale and compared the results with those derived from the D02 scale. Before we come to our conclusions, some information related to your concerns is added and presented first.

• The main objective of this study is to evaluate the effect of WRF setups on the precipitation simulations during the sub-daily heavy rainfall event on July 21st, 2012 in Beijing, China (sub-daily, regional scale).

• Beijing is the area of interest where the convective processes (convective-scale) happened, while the synoptic features (larger-scale) triggered the event, e.g., the eastward-moving vortex and the source of water vapor, were more spread outside this range. Therefore, 3 nested domains (D01, D02, D03) were set up, with D01 being the largest in the coarsest resolution, which covers the leading synoptic features, and D03 the smallest, which covers the area of interest (i.e., Beijing region).

• The initial conditions of the 3 domains were all provided by the ERA-Interim reanalysis and the regional geographical dataset. As for the lateral boundary conditions, D01 was forced by the ERA-Interim reanalysis, while D02 by D01, and D03 by D02.

• For the analysis/evaluation, the rainfall fields from D03 were mapped back onto the grid of the D02 domain. The analysis and evaluation were then done with the hybrid dataset D02+partly D03, on the D02 domain, against the ground rainfall observations and the ERA-interim reanalysis.

• The reasons for doing so are that (1) the spatial resolution of the reference truth (an hourly gridded dataset publicly available with the spatial resolution of 0.1 degrees) is commensurate with the D02 resolution, (2) the effect of some WRF model configurations (e.g., the domain size) on simulating this heavy rainfall event could not be well presented if it is just evaluated on the D03 scale.

In order to verify whether the choice of the evaluation domain is reasonable, we have recalculated all the metrics on the D03 scale by using another 3-hourly gridded dataset with a finer resolution of 0.05 degrees (Huang et al., 2013). A detailed comparison was then made based on the results derived from the two different scales (the D02 and D03 scale).

*Huang, C., Zheng, X., Tait, A., Dai, Y., Yang, C., Chen, Z., Li, T., and Wang Z.: On using smoothing spline and residual correction to fuse rain gauge observations and remote sensing data, J. Hydrol., 508, 410–417, 2013.*

By comparison, we have noticed that most of the evaluated results on the D03 scale were similar to those got on the D02 scale (See **Fig. 1** to **Fig. 8**). This has indicated that the experiment performs well on the larger scale could also have a good performance on the finer scale. It makes sense as the one with higher similarity to the larger-scale synoptic features tends to provide more accurate boundary conditions for the modeling in the inner domain. This, in other words, means that the experiment with good performance in the inner domain should also perform well in the larger-scale domain, which could be useful in evaluating the regional weather forecasts.

**[Figure 1 to Figure 8]**

Taking the domain size scenario as example, when it was evaluated on the D03 scale (See **Fig.1**), Case 0 with the smallest domain size performed better than the other two experiments in terms of the accuracy of rainfall during the first 18 hours. But from the D02 scale (See **Fig.2** and **Fig. 9**), it could be noticed that either the moving speed of the rain-belt or the magnitude of the maximum precipitation simulated by Case 0 was much different from the reference truth.

**[Figure 9]**

However, we agree that since our goal is to evaluate the effect of the WRF configurations on the heavy rainfall process in Beijing region. The choice of the hybrid domain for evaluation could lead to the possible ambiguity on distinguishing the source of the effect. Therefore, we would like to adopt the first option as suggested by you: repeating the evaluation on the D03 scale. As we have already recalculated the metrics for all the experiments on this scale, we are confident that 2-3 months are sufficient for us to revise this paper.

We appreciate your help in improving this manuscript, and we hope that our replies have addressed your remaining concerns.

Kind Regards!

The Authors

[Figure]

**Figure 1: Spatial values of the verification metrics calculated on the D03 scale for the experiments with different domain size (Case 0 has the smallest domain size covering Northern-Central China; Case 1 has the intermediate domain size covering Northern China and a part of the Mongol Country; Case 2 has the largest domain size covering the northeastern hemisphere; the statistic time durations are 6 h, 12 h, 18 h, and 24 h, respectively, counting from 12 am 21 July 2012).**

[Figure]

**Figure 2: As in Fig. 1, but the metrics were calculated on the D02 scale.**

[Figure]

**Figure 3: Spatial values of the verification metrics calculated on the D03 scale for the experiments with different vertical resolution (Case 1 has 29 vertical levels equal to that of the ERA-Interim reanalysis, Case 3 and Case 4 has doubled and tripled vertical levels, respectively).**

[Figure]

**Figure 4: As in Fig. 3, but the metrics were calculated on the D02 scale.**

[Figure]

**Figure 5: Spatial values of the verification metrics calculated on the D03 scale for the experiments with different horizontal resolution (Case 3 has the initial downscaling ratio of 1:3:3 with the outermost grid size of 40.5 km; Case 5 and Case 6 have the same outermost grid size with 1:5:5 and 1:7:7 downscaling ratio, respectively).**

[Figure]

**Figure 6: As in Fig. 5, but the metrics were calculated on the D02 scale.**

[Figure]

**Figure 7: Spatial values of the verification metrics calculated within 18 h time periods on the D03 scale for the WRF spin-up experiments (Case 5 has the initial spin-up time of 12 h; Case 7 is designed with 0 h spin-up time; From Case 8 to Case 18, the spin-up time is increased by 12 h from 24 h).**

[Figure]

**Figure 8: As in Fig. 7, but the metrics were calculated on the D02 scale.**

[Figure]

**Figure 9: Spatial distribution of six hour accumulated precipitation for the domain size experiments within the D02 domain during the heavy rainfall event from 12am, July 21, 2012 (a) precipitation of Case 0 (with the smallest domain size) in the first 6h; (b) precipitation of Case 0 in the second 6h; (c) precipitation of Case 0 in the third 6h; (d) precipitation of Case 1 (with the medium domain size) in the first 6h; (e) precipitation of Case 1 in the second 6h, (f) precipitation of Case 1 in the third 6h; g) precipitation of Case 2 (with the largest domain size) in the first 6h; h) precipitation of Case 2 in the second 6h; and i) precipitation of Case 2 in the third 6h.**

---

## Author Response (AR1)

Dear Associate Editor/Reviewers,

We would like to thank the editor and the referees for the helpful comments and thorough suggestions that surely help us improve the manuscript. We have made corrections according to these comments and marked the amended portion in blue in the annotated version of the manuscript. Below, we provide our point-to-point response to the comments and presented the main corrections in the revised manuscript.

**Response to the reviewer's comments**

*Reviewer #1*

"The authors present a study to investigate the impact of different domain sizes, vertical resolution, nesting ratios and spin-up time on a heavy precipitation event over Beijing. The simulations were forced by ERA-Interim reanalysis data available on 0.75 °resolution in six hourly intervals. The different experiments were performed using three domains with a two-way nesting approach and the innermost domain centered on Beijing. Sub-daily precipitation of the second domain was verified against gridded precipitation observations from the China Meteorological Center. In addition, the precipitable water content of the WRF simulations was validated with ERA-Interim reanalysis data as a proxy for the maximum possible precipitation.

In a general sense, this type of experiments is of great relevance for flash flood forecasting and early warning systems. However, in the current experimental setup, I see several critical points preventing the traceability of the results."

**Response**: Thank you for pointing this out. We really appreciate your time and effort invested in the review. **We have carefully checked all the issues and added the missing information to increase the traceability of the results.**

**General Comments**

**1.** - The authors apply 2-way nesting from the outer to the inner domains. This means that precipitation patterns of the 3rd domain (which is not analyzed in your study) are reflected in the second domain. This actually means that you verify the precipitation from domain three mixed with terrain and land use data from domain two. Why did you verify domain two instead of domain three in your study? When looking at Fig. 1, Beijing has complex terrain which is not accurately represented at 13.5 km resolution.

**Response**: Thank you for pointing this out. Indeed, Domain three covers the main convective process and should be the best choice for an urban flooding study. As you suggested, **we recalculated the metrics for all the experiments over domain three (D03)** by comparing the simulations with 3 hourly 0.05-degree gridded dataset, produced by fusing rain gauge observations and the CMORPH data (Huang et al., 2013). The evaluation was then repeated on the scale of D03.

**Detailed results can be seen in Section 5.** Besides, the evaluation made over the domain two was also used, but as an auxiliary method for subjective verification. This is based on the assumption that an experiment with good performance in the inner domain should also capture the large-scale features in the outer domain, as the appropriate representation of these large-scale features will result in more accurate boundary conditions.

Huang, C., Zheng, X., Tait, A., Dai, Y., Yang, C., Chen, Z., Li, T., and Wang Z.: On using smoothing spline and residual correction to fuse rain gauge observations and remote sensing data, *J. Hydrol.,* **508**, 410–417, 2013.

**2.** - Also with a 2-way nesting approach, you do not balance any kind of model physics with respect to the lateral boundary conditions. In a 2-way nesting approach, the fine grid resolution replaces the coarser scale resolution over the area of domain three.

**Response**: Thank you for pointing it out. We are sorry for not carefully checking this statement. Indeed, in a two-way nesting approach, the parent domain has its independent run, but it serves the child domain's boundary condition at each time step. And then after the child domain's dynamic modeling run, the child domain result (including the rainfall field) is mapped onto the target parent domain grids. **We have removed this confusing sentence in the revised manuscript.**

**3.** - The authors decided to use ERA-Interim reanalysis data to initialize their model simulations. As mentioned on page four, the resolution is $0.75°$. I am not sure if such a coarse resolution is able to provide reasonable initial conditions, especially when focusing on sub-daily rainfall.

**Response**: Thank you for raising this question. We agree that this dataset may not be the best choice if other sources of input fields with higher resolution were available. However, the ERA-Interim reanalysis dataset has been widely used in downscaling studies with acceptable results, and it is the best source of the reanalysis data accessible in the study area.

**4.** - Although you mention that a small domain may benefit from the lateral boundary conditions, I doubt that such a small outer domain of effectively 30*30 ($40\times40$) grid cells (due to boundaries of at least 5 cells in each direction) is sufficient here. This is also mentioned on page five in your manuscript. If you carefully checked the WRF webpage, you may have noticed that at least 100*100 cells are recommended for every domain.

**Response**: Thank you for pointing it out. In WRF-ARW, at least five cells along the boundaries of each domain are indeed required to mitigate sharp gradients (i.e., short wavelength features) that may exist along the lateral boundaries where the specified lateral boundary conditions differ from their values on the interior of the limited-area domain. As you pointed this out, we carefully rechecked our settings of all the experiments to make sure that we followed this recommendation. However, in our smallest domain size experiment (C0), eight cells were set between the outermost grid and the interim grid, and eleven cells were set between the interim grid and the innermost grid. We believe that these distances are sufficient for relaxation.

As for the domain size of at least $100 \times 100$, this should be related to the weather system in the study area. As long as the domain size covers the key weather features, it should be fine with a smaller domain than $100 \times 100$. Examples can be seen in Figure 3c of Bukovsky and Karoly (2009) where $45 \times 70$ cells were adopted in the outer domain covering the U.S. and in Figure 1a of Dasari et al. (2014) where $98 \times 55$ cells were used in the outer domain covering the most Europe. In our initial case (C0), the main feature of the weather systems that led to this storm was included within the outermost domain. Besides, to estimate the potential influence of lateral boundary conditions to the rainfall outputs, two other cases with larger domain size were designed for comparison.

Bukovsky, M. S., and Karoly, D. J.: Precipitation simulations using WRF as a nested regional climate model. *J. Appl. Meteorol. Climatol.*, **48(10)**, 2152-2159, doi:10.1175/2009JAMC2186.1, 2009.

Dasari, H. P., Salgado, R., Perdigao, J., and Challa, V. S.: A regional climate simulation study using WRF-ARW model over Europe and evaluation for extreme temperature weather events. *Intl. J. Atmospheric Sci.*, **9**, 2101-2122, doi:10.1155/2014/704079, 2014.

**5.** - It is also not clear how the WRF model levels are distributed in your simulation. From table 1, I only see that you used 29 levels up to 50 hPa. If a constant grid spacing of 1 km is applied, your model simulations will fail because processes in the PBL are not at all resolved. If you are in the middle troposphere, this spacing can be sufficient. Also the WRF tutorial and website suggest a vertical grid spacing of less than 1km. If you look at the user guide (e.g. http://www2.mmm.ucar.edu/wrf/users/docs/user guide V3.8/users guide chap5.htm#examples) you will see that at least ~40 levels are recommended when the model top is set to 50 hPa.

**Response**: Thank you for your suggestion. We agree that the experiment with slightly higher vertical resolutions could get better performance than the one with lower vertical resolution. This is also verified in our study where the experiment with 57 vertical levels (C3) shows better performance than the one with 29 vertical levels (C1). In this study, the ERA-Interim 29 pressure level data was selected as the initial forcing for two reasons. First, it meets the requirements of less than 1 km distance between each vertical level in the free troposphere where the convective processes mainly happen (See **Table 1**). Second, the NWP models used by the Chinese Meteorological Centre mainly employ 31 vertical levels in regional forecasting (See Table 11-2.1 in WMO, 2013).

**Table 1. The vertical levels set in the initial experiment and the corresponding height in Beijing, China.**

| Eta Level | 1 | 2 | 3 | 4 | 5 | 6 | 7 | 8 | 9 | 10 | 11 | 12 | 13 | 14 | 15 |
|---|---|---|---|---|---|---|---|---|---|---|---|---|---|---|---|
| Pressure (hpa) | 1000 | 975 | 950 | 925 | 900 | 875 | 850 | 825 | **800** | 775 | 750 | 700 | 650 | 600 | 550 |

| Eta value (0-1) | 1 | 0.973 | 0.947 | 0.921 | 0.894 | 0.868 | 0.842 | 0.815 | 0.789 | 0.763 | 0.736 | 0.684 | 0.631 | 0.578 | 0.526 |
|---|---|---|---|---|---|---|---|---|---|---|---|---|---|---|---|
| Height (km) | 0 | | | | 0.988 | | 1.457 | | 1.949 | | | 3.012 | | 4.206 | |

| Eta Level | 16 | 17 | 18 | 19 | 20 | 21 | 22 | 23 | 24 | 25 | 26 | 27 | 28 | 29 |
|---|---|---|---|---|---|---|---|---|---|---|---|---|---|---|
| Pressure (hpa) | 500 | 450 | 400 | 350 | 300 | **250** | 225 | 200 | 175 | 150 | 125 | 100 | 70 | 50 |
| Eta Value (0-1) | | | | | | | | | | | | | | |
| Height (km) | 5.574 | | 7.185 | | 9.164 | 10.363 | 11.784 | | | | | 16.18 | 18.442 | 20.576 |

Here, 1000pha-800pha corresponds to the lower troposphere; 800pha-250pha corresponds to the middle and upper troposphere. The approximation of the height was provided by a ground study in Beijing during the summer. However, although the ERA-Interim 29 pressure level data meets the requirement when simulating the heavy rainfall event in Beijing, it doesn't mean that it satisfies the conditions in other regions. **To illustrate this issue, one experiment forced with ERA-Interim 38 pressure level data (C5) has been added in S2 for comparison.** The results showed that C5 shows either better or worse performance than C1 in each period but produces less accurate rainfall simulations than C3 over most of the evaluated durations.

WMO: Anticipated advances in numerical weather prediction, and the growing technology gap in weather forecast. 2013. [Available online at https://www.wmo.int/pages/prog/www/swfdp/Meetings/documents/Advances_NWP.pdf]

**6.** - As your outer domain gets enlarged towards the pole, how did you deal with the map factors?

**Response**: Thank you for pointing it out. In this study, our main focus is on the simulation in domain two and domain three which are located in the medium latitude of the northern hemisphere, so the Lambert conformal projection was selected and centered on the same latitude (42.25°N) and longitude (114.0°E).

[Figure]

**Figure 1. The relative location of the nested domains adopted in C2 that with the largest outer domain of all the experiments.**

**Figure 1** shows the domain configuration with the largest outer domain. In this figure, it can be seen that the boundary of the outer domain is still far away from the pole and centered at the medium latitude area. **The above-mentioned information has been added to the revised manuscript. Please see Page 8, Line 22 – 25.**

**7.** - Choosing an adaptive time step may save computation time but is not the best way for scientific experiments (see WRF webpage).

**Response**: Thank you for raising this problem. As you mentioned, choosing an adaptive time step may not be the best way for scientific experiments. We are sorry for not carefully checking this statement and giving the impression that all the experiments in this study adopt the adaptive time step. In fact, C4 with the finest vertical resolution used this setting, as well as C6 with the finest horizontal resolution. When running these two cases, the recommended minimum time step was set (about $3 \times$ DX seconds for the outermost domain), but instability was encountered, and the model ran much slower than we expected and stopped before the end time. To deal with this problem, an adaptive time step was adopted, where the maximum time step was up to $6 \times$ DX with CFL value set to 1.2. **We have removed this confusing sentence in the revised manuscript.**

**8.** - The rescaling of the error measures may lead to a misinterpretation. In case of POD, how did you choose the factor 0.115? This is not clear from the manuscript.

**Response**: Thank you for pointing it out. It is our negligence to forget adding the related reference to illustrate how the scale for each verification parameter was selected. Here, we add the reference in which the same method was adopted to allow a convenient and multidimensional assessment of precipitation simulation quality for various WRF configurations. Please see Table 2 in Sikder and Hossain (2016).

In the case of POD, the factor of 0.115 was determined by the largest POD value calculated from all the experiments. Then all the POD values were divided by this factor to ensure they fell within the range of 0-1. The detailed illustration and reference has been added to the revised paper. **Please see Page 11, Line 10 - 11.**

Sikder, S., and Hossain, F.: Assessment of the weather research and forecasting model generalized parameterization schemes for the advancement of precipitation forecasting in monsoon-driven river basins, *J. Adv. Model. Earth Syst.*, **8**, 1210–1228, doi:10.1002/2016MS000678, 2016.

**9.** - Is the maximum RMSE used for each individual time step or is it calculated from an average over all the time steps?

**Response**: Thank you for raising this problem. The RMSE was calculated through the same process as POD. The procedure has been illustrated by Tian et al. (2016) in comparing the simulations to the observations from the ground meteorological stations. As the temporal duration was shorter and the spatial calculation was at grid scale, we merely mentioned the difference when we adopted the method. In this study, all the metrics were firstly computed between the observations and simulations of the same grid at each time step and then averaged within different time durations (6h, 12h,

18h, 24h) for the final analysis. **The related information has been added into the revised manuscript. Please see Page 10, Line 23 – 27.**

Tian, J. Y., Liu, J., Li, C. Z., and Yu, F. L.: Numerical rainfall simulation with different spatial and temporal evenness by using WRF multi-physics ensembles. *Nat. Hazards Earth Syst. Sci.*, **17**, 563-579, doi:10.5194/nhess-17-563-2017, 2017.

**10.** - It is also hard to believe that the POD remains constant, independent whether you start one week or 12 hours before the event?

**Response**: Thank you for raising this problem. In our study, as the POD was averaged among the large domain with at least $72 \times 72$ grid. The differences presented could be less obvious than expected. Besides, when shown in the same graph with the other parameters, the differences were much less obvious than others. But it doesn't mean it is constant.

**11.** - Also, what is the precipitation threshold used to calculate POD? Is it 0.1mm? Usually, POD is applied for different thresholds.

**Response**: Thank you for raising this idea. In this study, 0.1 mm is used to calculate POD. POD with different thresholds may be useful to investigate the accurate hit of the heavy rainfall area further. Considering that RMSE and R could also reflect this feature, we only choose POD with 0.1mm as one of the verification parameters in this study. **We have added the information into the revised version to make it clear. Please see Page 10, Line 16 – 17.**

**12.** - How did you match both grids together? Did you use CDO, NCO, or NCL for this? It seems that you applied a $1/R^2$ approach to remap the CMC precipitation observations to the WRF grid. What is the radius of influence in this case? This can strongly determine the resulting field, especially in case of heavy and localized precipitation.

**Response**: Thank you for pointing it out. We are sorry to miss the detailed illustration of the interpolation process. Here, for the rainfall values, the bi-linear interpolation method was adopted to remap the WRF simulations to the reference grid. In this method, four nearest points for each WRF grid were searched to accomplish the bi-linear interpolation process. In WRF-ARW, the bi-linear interpolation method is used as the default choice to interpolate the initial meteorological fields. **The related illustration has been added in the revised manuscript in Page 10, Line 14 - 15.**

**13.** - Have the integrated water vapor fields been handled in the same way?

**Response**: Thank you for raising this question. The initial water vapor field was extracted from WPS outputs by adding TCWV (Total column water vapor) in the Vtable files. This means that it has the same location as the field calculated from the WRF outputs. Therefore, there is no need to remap one field to another. **The related illustration has been added in the revised manuscript. Please see Page 10, Line 22 - 23.**

**14.** - It would be very useful, if the authors provide horizontal plots of the precipitation patterns to substantiate the results. The applied scores do not necessarily tell if the precipitation is simulated spatially correctly.

**Response**: Thank you for your suggestion. The plots of the spatial precipitation patterns indeed would be helpful to substantiate the results, such as the examples of domain size experiments. **Some representative plots have been added to the revised manuscript to make the results with more clarity. Please see Page 12, Line 29 – 30 to Page 13, Line 1 - 11.**

"In my opinion, a lot of important information is missing here and I also see deficiencies in the experimental setup. I strongly suggest that a native English speaker reads through the manuscript."

**Response**: We hope our replies have addressed your concerns. **The revised manuscript has been thoroughly proof-read by a native English speaker.**

*Reviewer #2*

"In this contribution, the authors evaluate the performance of the WRF model in different configurations for a single heavy rainfall event centered over Beijing, China. The evaluation differs from other studies in that field in the sense that no physics parameterization evaluation is attempted. Instead, the model setup (domain configuration, number of vertical levels = vertical resolution, nesting ratio = horizontal resolution, forecasting lead time) are explored. In the order of the above, the best configuration is chosen in each step to perform several experiments in the next step. Several verification measures are employed for precipitation and precipitable water (PW).

The design of these experiments is convincing despite a few weak points listed below. The use of English language, however, needs improvement. Grammatical mistakes and strange wordings render some parts of the text unclear. I did not make any attempt to correct this but highlight a few common issues below. With improvements to the language and several changes to the contents, the contribution may be suitable for publication."

**Response**: Thanks very much for the encouraging feedback.

**General remarks**

**1.** - Dependence of parameters varied: Although discussed in the introduction, the dependence of optimal lead forecasting time on domain extent (and vice versa) is not considered in the study. Instead, based on a standard lead time of 12h, the

"best" domain configuration is derived as C1, based on which an optimal lead time of 60h is inferred later on as C11. In my understanding, these two lead times should match if one really found the "best" combination of these two parameters.

**Response**: We agree that there is a dependence relationship between domain extent and forecasting lead time and this is particularly true in the limited-area modeling cases where data assimilation is not conducted. For instance, a model run with a larger domain size may need longer lead time to spin-up physical processes of interest, such as clouds, precipitation, local ageostrophic circulations, and lateral boundary conditions. Besides, the choice of lead time and domain size at the same time determine the moment at which the initial and lateral boundary conditions are derived and the range of the corresponding synoptic features and water vapor conditions involved at that moment. This, however, means that for a given domain size, the results may also be affected by the degree of similarity between the forcing data and the real conditions at the initialization time. Besides, the choice of updating lateral boundary conditions at a given interval could affect the time needed to spin-up the physical processes. It is noteworthy that although C1 is detected with the best performance, C2 with nearly doubled domain extent of C1 only differs obviously in the PW fields but with less diversity regarding distribution characteristics of the rain belt. Combining all the aforementioned factors, we believe that it makes sense that C11 (with 60h lead time) is evaluated with better performance when compared with C5 (with 12h lead time).

**2.** - The authors choose an adaptive time step to conduct their modeling experiments. This introduces another free parameter, since the actual time step adopted in each simulation may vary and as such influence the results.

**Response**: As you mentioned, choosing an adaptive time step may introduce another free parameter and as such influence the results. We are sorry for not carefully checking the statement and giving the impression that all experiments adopted the adaptive time step. In fact, only C4 with the finest vertical resolution, and C6 with the finest horizontal resolution used this setting. When running these two cases, the recommended minimum time step was set (about $3\times DX$ seconds for the outermost domain), but instability was encountered, and the model ran much slower than we expected and stopped before the end time. To deal with this problem, an adaptive time step was adopted, where the maximum time step was up to $6\times DX$ with CFL value set to 1.2. **We have removed this confusing statement in the revised manuscript.**

**3.** - The performance evaluation of the WRF model is performed over the intermediate domain D02 and not over the highest-resolution domain D03. It is argued that the two-way nesting approach does inform D02 about the results in the innermost domain D03, but interpolation to the coarser D02 grid and the (possible) difference in setup of the model physics (see next point) may influence the conclusions drawn. I would like to encourage the authors to conduct simulations without D03 for their best setup at least.

**Response**: Thank you for raising this question. Indeed, domain D03 covers the main convective processes and should be the best choice for evaluation. **As you suggested, we have recalculated the metrics for all the experiments over**

**domain three (D03) and repeated the evaluation on the scale of D03. The analyzed results can be seen in Section 5 of**

**the revised manuscript.**

**4.** - Model physics: It is unclear to me whether the GD cumulus parameterization is also employed in D03 at convection-permitting resolution (<5-10km).

**Response**: Thank you for raising this question. In this study, GD cumulus parameterization was turned on for each domain, including D03 at the convection-permitting resolution between 1km and 5km, to represent the effects of sub-grid scale convective processes which were also detected in the rainfall processes of this heavy rainfall event. **We have added the**

**information into the revised manuscript to avoid the confusion. Please see Page 8, Line 23 - 25.**

**5.** - The forcing data is obtained from ERA-Interim on pressure levels (28 (29) levels). This is not ideal, in particular since the authors are trying to assess the added value of a higher vertical resolution and since ERA-Interim is also available on

38 model levels. I would like to encourage the authors to repeat experiments with their optimal setup C11 using

ERA-Interim model-level data and varying the vertical resolution as in S2, for example.

**Response**: Thank you for your suggestion. We agree that the experiment with slightly higher vertical resolutions could get better performance than the one with lower vertical resolution. This is also verified in this study where the experiment with

57 vertical levels (C3) shows better performance than the one with 29 vertical levels (C1). In this study, the ERA-Interim

29 pressure level data was selected as the initial forcing for two reasons. First, it meets the requirements of less than 1 km distance between each vertical level in the free troposphere where convective processes mainly happen (See **Table 1**).

Second, the NWP models used by the Chinese Meteorological Center are mainly operated with 31 vertical levels for regional forecasting (See Table 11-2.1 in WMO, 2013).

**Table 1. The vertical levels set in the initial experiment and the corresponding height in Beijing, China.**

| Eta Level | 1 | 2 | 3 | 4 | 5 | 6 | 7 | 8 | 9 | 10 | 11 | 12 | 13 | 14 | 15 |
|---|---|---|---|---|---|---|---|---|---|---|---|---|---|---|---|
| Pressure (hpa) | 1000 | 975 | 950 | 925 | 900 | 875 | 850 | 825 | **800** | 775 | 750 | 700 | 650 | 600 | 550 |
| Eta value (0-1) | 1 | 0.973 | 0.947 | 0.921 | 0.894 | 0.868 | 0.842 | 0.815 | 0.789 | 0.763 | 0.736 | 0.684 | 0.631 | 0.578 | 0.526 |
| Height (km) | 0 | | | | 0.988 | | 1.457 | | 1.949 | | | 3.012 | | 4.206 | |

| Eta Level | 16 | 17 | 18 | 19 | 20 | 21 | 22 | 23 | 24 | 25 | 26 | 27 | 28 | 29 |
|---|---|---|---|---|---|---|---|---|---|---|---|---|---|---|
| Pressure (hpa) | 500 | 450 | 400 | 350 | 300 | **250** | 225 | 200 | 175 | 150 | 125 | 100 | 70 | 50 |
| Eta Value (0-1) | | | | | | | | | | | | | | |
| Height (km) | 5.574 | | 7.185 | | 9.164 | 10.363 | 11.784 | | | | | 16.18 | 18.442 | 20.576 |

However, although the ERA-Interim 29 pressure level data meets the requirement when simulating the heavy rainfall event in Beijing, it doesn't mean that it satisfies the condition in other regions. **To illustrate this issue, one experiment forced**

**with ERA-Interim 38 pressure level data (C5) has been added in S2 for comparison.** The results shows that C5 has either better or worse performance than C1 in each analyzed period, but is verified with less accuracy in the rainfall simulations than C3 throughout most of the evaluated time durations.

WMO: Anticipated advances in numerical weather prediction, and the growing technology gap in weather forecast.

2013.[Available online at https://www.wmo.int/pages/prog/www/swfdp/Meetings/documents/Advances_NWP.pdf]

**6.** - Several of the abbreviations in the text or the figure captions are not introduced before they are used (or not at all), please check and correct.

**Response**: Thank you for pointing it out. **We have checked all the abbreviations in the text and figure captions to**

**make sure their full names are given before they are used. Please see the figure captions in Page 24.**

**7.** - The statistical measures used here have different directions of "good". (i.e. RMSE is good if low, R is good if high).

This is not mentioned anywhere in the text and in the figures, which makes the interpretation confusing, also because the statistics are rescaled. I would encourage to state explicitly what value implies a better model performance for which statistical measure, and possibly encode this (in color or differently) in the figures and the tables.

**Response**: Thank you for raising this question. To make it clear, the verification metrics have been assigned with a new set of symbols after the statistics are rescaled (see **Table 2**). **The explanation of the performance measures has been added**

**in the revised manuscript as suggested. Please see Page 11, Line 7 - 14.**

**Table 2. Correlations between original and rescaled verification metrics.**

| Original Metrics | Representative Meaning | Rescaled Metrics | Threshold Value |
|---|---|---|---|
| $POD$ | Probability of Detection | $POD' = POD/POD_{max}$ | + 0.115 max |
| $RMSE$ | Root Mean Square Error | $RMSE' = 1 - RMSE/RMSE_{max}$ | + 41 max |
| $R$ | Pearson Correlation Coefficients | $R' = R$ | N/A |
| $WRMSE$ | RMSE of the Precipitable Water | $WRMSE' = 1 - WRMSE/WRMSE_{max}$ | + 7.3 max |
| $WR$ | R of the Precipitable Water | $WR' = WR$ | N/A |
| $RE_{PMAX}$ | Relative Error of the Maximum Precipitation | $PMAX' = RE_{PMAX}$ | N/A |
| $RE_{TP}$ | Relative Error of the Total Precipitation | $TP' = RE_{TP}$ | N/A |

**8.** - Beneath dynamical downscaling explored here, also statistical downscaling methods and new global modeling approaches on irregular grids (e.g. MPAS) have been used and show promising results to forecast extreme precipitation events. This should be discussed briefly in the introduction or discussion section.

**Response**: Thank you for your suggestion. We agree that some statistical downscaling methods, along with data assimilation methods, could provide more reliable forecasts of extreme precipitation events, especially for short-term forecasting with 6h-24h lead time. **The related content has been added in the discussion section as suggested. Please see Page 3, Line 1 - 3, Line 10 – 11 and Page 17, Line 20 – 23.**

**Specific remarks**

**1.** - Page 2, lines 24-26: WRF being used at resolutions >10km only is not true. Many leading operational NWP centers are employing WRF at convection-resolving resolution (NCEP: HRRR, 3km over CONUS; Meteo Group: 3km over Europe; New Zealand Met Service: <4km over New Zealand) operationally.

**Response**: Thank you for the clarification. **We have rephrased this sentence in the manuscript to avoid the confusion. Please see Page 2, Line 27 – 30.**

**2.** - Page 4, line 27: Isn't ERA-Interim available from 1979 (not 1989)?

**Response**: Thank you for pointing this out. Indeed, ERA-Interim is available from 1979 (originally, ERA-Interim ran from 1989, but the 10-year extension for 1979-1988 was added in 2011. **We have amended this statement. Please see Page 5, Line 1.**

3. - Page 7, line 31: RRTMG schemes (not RRTM)? Or is it "RRTM" for LW and "Dhudia" for SW?

**Response**: Thank you for pointing this out. The radiation schemes adopted in our study were the RRTMG schemes. **We have edited this sentence** to: "The radiation processes are represented by the RRTMG shortwave radiation and the RRTMG longwave radiation schemes (Iacono et al., 2008)." **Please see Page 8, Line 10 - 11.**

**4.** - Page 14, lines 2-15: the discussion is confusing for the reader as he/she is expected to translate nesting ratios into effective horizontal resolution. In this paragraph, as well as in Table 1, the effective horizontal resolution should be stated explicitly, alongside with the nesting ratios.

**Response**: Thank you for the suggestion. **We have followed this advice, as noted above. Please see Page 14, Line 23 – 24 and Page 33, Table 1.**

**5.** - Page 14, line 31 to page 15, line 1: the authors state that the positive bias in precipw (PMAX) depends on the initialization time, with largest biases for initialization times with highest amounts of precipw. This, in my opinion, is an important finding and should be highlighted and possibly discussed further.

**Response**: Thanks for your suggestion. **We have highlighted this finding and discussed it as suggested. Please see Page 15, Line 27 – 30.**

**6.** - Page 17, lines 8-11: The authors briefly discuss the dependence on the quality of the forcing data. This highlights the importance to conduct additional experiments with ERA-Interim model level data as described above, and at least discuss (if not evaluate) potential effects when using ECMWF high-res forecasts on 137 model levels and approx. 9km horizontal resolution.

**Response**: Thank you for pointing this out. As mentioned above, one experiment forced with ERA-Interim 38 model level data (C5) has been added in Scenario two for comparison. Besides, the possible effect of the quality of the forcing data on the forecasts has been briefly discussed in the revised manuscript. **Please see Page 17, Line 23 – 25.**

**Typographical corrections**

**1**. - Page 4, line 10: "coaster-scale" -> "coarser-scale"

**2**. - Page 4, line 21: "Earth-system system"

**3**. - Page 4, line 29: "WRFV3.7.2" or "WRFV3.7.1"

**4**. - Page 5, line 14: "They two together" -> "The two together""

**5**. - Page 5, line 6: the correct reference should be Skamarock et al. (2008).

**6**. - Page 13, line 11: "less sensititvity" -> "less sensitive"

**Response**: Thanks for reading our manuscript so meticulously – **these have been corrected**.

**Grammar corrections**

**1**. - Language-specific: - the word "occurred" is often missing a leading "that" or "which" -the expression "demonstrated true" seems odd to me - several times in the text, "grid" is used whereas "grid point" should be - singular and plural, as well as the use of articles need to be checked carefully.

**2**. - Example for a necessary rewording: caption of figure 1: "Location and topography of the study area. Left panel: three levels of nested domains adopted in most experiments, with D03 covering the Beijing area; right panel: zoom-in on the topography of the Beijing area".

**Response:** Thank you for pointing this out. **We have carefully checked through the manuscript to correct the grammatical mistakes with more precise descriptions**.

We hope our replies have addressed your concerns, and **the revised manuscript has been thoroughly proof-read by a native English speaker.**

**Response to the editor's comments:**

*Associate Editor*

**Specific Comment**

"- I have read the referees' comments and your replies. While for most points, your replies are satisfactory, I need some more information from you about a major point both referees raised: The choice of modeling and evaluation domains (comment 2 (General Comment **1**) of referee #1, comment 4 (General Remarks **3**) of referee #2). In the following, I will explain my current understanding of what you were doing, and my related conclusions. Before I take my decision about how to proceed, I would like to verify that I understood things correctly:"

**Response:** Thank you very much for your positive assessment and constructive suggestions to our study.

" •You model an extreme rainfall event using WRF set up on 3 nested domains (D01, D02, D03) with D01 being the largest domain in coarsest resolution, D03 the smallest, which covers the area of interest, Bejing region.

• When running the models, D01 is forced by ERA-Interim reanalysis, D02 by D01, D03 by D02

• For analysis/evaluation, you map rainfall from D03 back into the results and onto the grid of the D02 domain.

• Analysis and evaluation is the done with this hybrid data set D02+partlyD03, on the D02 domain, against ground rainfall observations and ERA-interim reanalysis.

• The reasons for doing so is that a) the reference truth is available only in a resolution comparable to the D02 resolution, and b) differences among the models are less obvious in the D03 domain than in the D02 domain."

**Response:** Thank you for reading our manuscript so meticulously. We agree that most of your understanding is correct.

Beijing is the area of interest where the convective processes (convective-scale) happened. Three nested domains (D01,

D02, D03) were set up, with D01 being the largest in the coarsest resolution, which covers the leading synoptic features, and D03 the smallest, which covers the Beijing region. When running the model, D01 is forced by ERA-Interim reanalysis,

D02 by D01, D03 by D02. In the original version of the manuscript, the analysis/evaluation is conducted on the D02 scale by comparing the hybrid dataset D02+partly D03 against ground rainfall observation. The reasons for doing so are that (1)

the spatial resolution of the reference truth (an hourly gridded dataset publicly available with the spatial resolution of 0.1

degrees) is commensurate with the D02 resolution, (2) the effect of some WRF model configurations (e.g., the domain size)

on simulating this heavy rainfall event could not be well presented if it is just evaluated on the D03 scale.

"So if this is correct, I have two main concerns:

• If your goal is to evaluate different WRF setups with respect to regional (here: D03 or Bejing-scale) heavy precipitation, then a) the evaluation should be done exactly on this scale and b) they should be compared to reference data with adequate resolution for that scale.

• If your goal is to evaluate different WRF setups with respect to larger-scale (here: D02-scale) precipitation patterns, then the D03 run on its small scale is unnecessary. You could then run WRF on the D02 domain with different configurations (the D02 and the D03 settings with respect to parameterization, grid size etc.) and compare these.

So if my concerns are based on correct understanding, you will either have to
• gather better reference data and repeat the evaluation on the D03 scale, or
• do new model runs on the D02 scale and repeat evaluation on the D02 scale."

**Response:** To verify whether our choice of the evaluation domain is reasonable, we recalculated all the metrics on the D03 scale by using another 3-hourly gridded dataset with a finer resolution of 0.05 degrees (Huang et al., 2013). The results were then compared with the results derived from the D02 scale.

By comparison, we have noticed that most of the evaluated results on the D03 scale were similar to those got on the D02 scale. This indicates that the experiment performs well on the larger scale could also have a good performance on the finer scale. It makes sense as the one with higher similarity to the larger-scale synoptic features tends to provide more accurate boundary conditions for the modeling in the inner domain. This, in other words, means that the experiment with good performance in the inner domain should also perform well in the larger-scale domain, which could be useful in evaluating the regional weather forecasts.

Taking the domain size scenario as example, when it was evaluated on the D03 scale (See **Fig. 2**), Case 0 with the smallest domain size performed better than the other two experiments in terms of the accuracy of rainfall during the first 18 hours. But from the D02 scale (See **Fig.3** and **Fig. 4**), it could be noticed that either the moving speed of the rain-belt or the magnitude of the maximum precipitation simulated by Case 0 was much different from the reference truth.

[Figure 2 to Figure 4]

However, we agree that since our goal is to evaluate the effect of the WRF configurations on the heavy rainfall process in Beijing region. The choice of the hybrid domain for evaluation could lead to the possible ambiguity on distinguishing the source of the effect. **Therefore, we have adopted the first option as suggested by you: repeating the evaluation on the D03 scale. Detailed results can be seen in Section 5. Meanwhile, the evaluation made on the D02 scale has also been adopted, but as an auxiliary method for subjective verification. Related results can be seen in Section 5.1 and Section 5.4.**

We really appreciate your help in improving this manuscript, and we hope that our replies have addressed your concerns.

Kind Regards,

The Authors

[Figure]

Figure 2: Spatial values of the verification metrics for the WRF domain size experiments, calculated over different temporal
durations and over domain three. Case 0 incorporates the smallest domain, which covers north-central China; Case 1
incorporates a domain of intermediate size that covers northern China and part of Mongolia; and Case 2 incorporates the
largest domain, which covers the Northeastern Hemisphere. The metrics are calculated over time periods of 6 h, 12 h, 18 h, and
24 h that begin at 12 am on 21 July 2012.

[Figure]

Figure 3: As in Fig. 1, but the metrics were calculated on the D02 scale.

[Figure]

**Figure 4: Spatial distribution of 6-h accumulated precipitation for the domain size experiments within domain two of Case 0**

**during the heavy rainfall event beginning at 12 am on July 21, 2012. (a) Precipitation in Case 0 (with the smallest domain size)**

**during the first 6-h period; (b) precipitation in Case 0 during the second 6-h period; (c) precipitation in Case 0 during the third**

**6-h period; (d) precipitation in Case 1 (with the medium-sized domain) during the first 6-h period; (e) precipitation in Case 1**

**during the second 6-h period; (f) precipitation in Case 1 during the third 6-h period; g) precipitation in Case 2 (with the largest**

**domain) during the first 6-h period; h) precipitation in Case 2 during the second 6-h period; and i) precipitation in Case 2**

**during the third 6-h period.**

**List of all relevant changes**

**New Title** "Evaluation of the ability of the Weather Research and Forecasting model to reproduce a sub-daily extreme rainfall event in Beijing, China using different domain configurations and spin-up times".

**Page 1, Line 4 and Line 6** one affiliation of the co-authors was added.

**Page 1, after Line 18** the original statements were replaced by "A comparison of the optimal run and the initial run performed using the most common settings reveals clear improvements in the verification metrics. Specifically, R'

increases from 0.226 to 0.67; the relative error of the maximum precipitation at a point increases from 0.44 to 0.883; and the cumulative spatial error decreases by 33.65 %."

**Page 2, after Line 12** we added:

"Recently developed statistically-based rainfall generation methods and remote sensing data have been shown to enable the extension of the lead time to 24 hours. However, this lead time is still insufficient to provide effective flood mitigation for medium or large urban areas with very short hydrologic response times (Shih et al., 2014; Li et al., 2017)."

The correction was made based on the General Remark 8 of Reviewer #2.

**Page 2, after Line 27** the original statements were replaced by:

"Increasing numbers of meteorological operational centres and research groups are adopting these new NWP models to carry out simulations of heavy rainfall events or real-time forecasting. The resolutions of the rainfall products have improved from tens of kilometres to less than a kilometre, and the lead times have increased from less than a day to more than a week (WMO, 2013)."

The correction was made based on the Specific Remark 1 of Reviewer #2.

**Page 2, after Line 30** we added:

"Meanwhile, case studies have been carried out using regional convective-resolving models to evaluate the local rainfall predictions generated by sophisticated regional nesting techniques or the global smooth grid transition approach on unstructured grids (Hong and Lee, 2009; Soares et al., 2012; Sikder and Hossain, 2016; Heinzeller et al. 2016)."

The correction was made based on the General Remark 8 of Reviewer #2.

**Page 3, after Line 10** we added:

"These uncertainties are expected to be further magnified by downscaling or the use of mesh transition procedures, so re-evaluation and calibration of the related model configurations are commonly required."

The correction was made based on the General Remark 8 of Reviewer #2.

1 **Page 4, Line 15** "a coaster-scale reanalysis" was replaced by "a coarser-resolution reanalysis".

2 **Page 4, Line 26** "an Earth-system system" was replaced by "an Earth system".

3 **Page 4, after Line 30** the original statement was replaced by:

4 "The final reanalysis product, ERA-Interim, is a global gridded dataset that is available at a spectral resolution of T255 and

5 at both the 60 levels used in the model and 38 interpolated pressure levels for all dates beginning on 1 January 1979

6 (Berrisford et al., 2009; Dee et al., 2011)."

7 The correction was based on the Specific Remark 2 of Reviewer #2.

8 **Page 5, after Line 2** we added:

9 "Here, the ERA-Interim pressure-level data are selected as the initial forcing. One reason is that, as is necessary, the

10 vertical grid spacing between the adjacent pressure layers is less than 1 km in the free troposphere, where the convective

11 processes mainly occurred during the Beijing SDHR event. In addition, the NWP models used by the Chinese

12 Meteorological Centre mainly employ 31 vertical levels in regional forecasting (WMO, 2013)."

13 The correction was made based on the General Comment 5 of Reviewer #1 and General Remark 5 of Reviewer #2.

14 **Page 5, Line 7** "WRF 3.7.2" was replaced by "The advanced WRF (ARW-WRF) model, version 3.7.1".

15 **Page 5, Line 14** the reference was amended to "Skamarock et al. (2008)".

16 **Page 5, Line 20** "they two together" was replaced by "Together".

17 **Page 8, after Line 10** the original statement was replaced by:

[revised manuscript text omitted]

The content was added based on the General Comment 8 of Reviewer #1.

**Page 11, after Line 20** the Results and Discussions (Section 5 and Section 6) were rewritten based on reevaluation of the performance of WRF model on the D03 scale by using a higher resolution reference dataset.

The corrections were mainly made based on the General Comment 1 of Reviewer #1 and the General Remark 3 of

Reviewer #2.

**Page 12, after Line 29** we added:

[revised manuscript text omitted]

**Figure and table captions:**

All the abbreviations in the figure and table captions were checked to make sure their full names are given before they are used. Besides, the captions were rewritten to avoid rewording.

This correction was made based on the General Remark 6 of Reviewer #2.

**Figures were amended:**

Figure 3, 5, 6, 7 were replaced by the subfigures which showed the spatial values of the verification metrics for the WRF

experiments calculated over domain three.

**New figure was added:**

Figure 4 was added, showing the spatial distribution of 6-h accumulated precipitation for the domain size experiments within domain two of Case 0 during the heavy rainfall event beginning at 12 am on July 21, 2012.

**Data in the tables were amended:**

The results in the tables were updated after we repeated the evaluation on the D03 scale.

**The marked-up manuscript version**

[revised manuscript text omitted]

---

## Referee Report (RR1)

**General Comments**

The authors present a study to investigate the impact of different domain sizes, vertical resolution, nesting ratios and spin-up time on a heavy precipitation event over Beijing. The simulations were forced by ERA-Interim reanalysis data available on 0.75° resolution in six hourly intervals. The different experiments were performed using three domains with a nesting approach while the innermost domain is centered on Beijing. Sub-daily precipitation of the third domain was verified against a merged precipitation product in 3 hourly intervals.

Generally, this type of study is of importance as domain size, vertical and horizontal resolution can either have a positive or detrimental impact on the forecast quality.

However, in the present manuscript, it is not really clear how the impact of the previously mentioned can be isolated. For me it is also still a major concern to investigate a sub-daily heavy rainfall event and drive the model with coarse resolution ERA-Interim data. I highly recommend to repeat at least one of the simulation by applying the high-resolution ECMWF operational analysis or ERA5 reanalysis data.

I also feel that the rescaled scores causes more confusion to the reader, rather than improving the readability.

Compared to the first draft, the English spelling and grammar is considerably improved.

The manuscript can be accepted after major revisions. Please the reasons below.

***Specific Comments:***

From the manuscript it is not clear to me, whether you applied 1-way or 2-way nesting. This information is missing for the reader.

The size of your innermost domain is never mentioned but this can play an important role. In case it's just 100*100 cells, you have to subtract 10 cells in both direction due to boundary effects so you will have an effective grid size of 90*90 levels and I have a feeling that this is not sufficient at this particular high resolution.

For a better readability, I suggest the following section ordering:

2 – Meteorological situation: The paragraph from section 3.1 on page 7 can be moved here.

3 Experimental design

3.1 WRF Model: The paragraph on page 5, lines 7-17 can be moved here.

3.2. Model configuration

  - Forcing: The paragraph on page 4, starting line 24 can be moved here.

  - Model setup end experiments: The text between page 8, lines 5 tp page 9, line 29 can be moved here.

Abstract: What do you mean with "cumulative spatial error"? I do not find any explanation in the manuscript.

Page 2, line 24: Which period?

Page 2, line 25: Please use Skamarock et al. (2008) as reference here.

Page 3, line 18: "relatively" repeated twice. I think "relative" is not appropriate here.

Page 3, line 21: I do not agree here. For sure there are studies about this topic.

Page 3, line 25: I guess you are talking about Beijing here.

Page 4, line 4: Did you use the setting from the two publications in your study? If this is the case, please mention this.

Page 4, line 6: What is the first question here?

Page 4, line 15: Reference to ERA-Interim? This paragraph can be splitted into two parts and the second half may be moved to section 4.

Page 5, line 1: I think Dee et al. (2011) is sufficient here.

Page 5, lines 4-5: What is the CMC model saying in this case? Is it better than WRF? I am also not confident that the CMC model configuration is the justification to apply coarse resolution input and boundary conditions.

Page 5, lines 19-29: Please check if the full paragraph is necessary here. The focus in your study is not evaluating LBCs.

Page 5, line 31: Do you really mean grid spacing here? Or do you mean the number of grid cells?

Page 6, lines 1-2: I think a vertical resolution of > 1 km in the PBL is far not sufficient here. I also do not agree with this statement in general.

Page 6, line 7: What is an excessive grid spacing?

Page 6, line 10: see my comment two lines above. I thinks this is wrong.

Page 6, line 18-20: Do you really mean spin-up times here? Or do you mean sth. like forecast lead times?

Page 6, line 25: Are you sure?

Page 7, line 15: I would rather write 60 m to 2300 m.

Page 7, line 22: You mention 100 mm/h rain rate, but your verification is performed over at least 6 hourly windows. How does this fit together?

Page 8, line 5: What is the reason you also apply the cumulus parametrization in D03? Please explain. I am not sure if the GD is designed to run at very high resolution. If you see the

necessity to use a convection parametrization in D03, why didn't you choose a more recent scheme like G3 or Grell-Freitas?

Page 8, line 10: I think Ek et al. is the wrong reference for the MM5 surface layer scheme.

Page 8, lines 22-25: I think this is not correct as you also change the spin-up period. There is no other choice than using the same map projection in a multi domain simulation.

Page 8, line 27: Please mention the recommended setting you refer to in section 2. What is the time step you applied? This is not mentioned in the manuscript? If an adaptive time step is applied, then the results are not comparable.

Page 8, line 29: The odd ratio is selected due to the applied Arakawa grid and not because of a reduction of the initial error.

Page 9, lines 2-3: This should be clear.

Page 9, line 6: Why did you output the D03 data only in 3 hourly intervals? Is it due to the merged CMORPH data set? Why didn't you use the hourly gridded precipitation data set from the CMC? Especially when you are interested in sub-daily extreme rainfall?

Page 9, lines 11-13: This implies that your simulation C0 is unusable here as it does not capture the situation. What is a perturbed synoptic feature?

Page 9, line 16: This is totally confusing. D02 of the experiment C0 is now the new D01 for C1? I do not see the equivalence here: 72*72 cells at 13.5 km vs. 80*64 cells at 40.5km resolution.

Page 9, line 22: I guess you mean the middle troposphere. Why is this condition not satisfied in other regions?

Page 9, line 25: I think there is no "increased nesting ratio" of 1:3:3.

Page 10, line 6: ERA-Interim provides precipitation on a 6 hourly basis.

Page 10, line 12: What do you mean with "scale of D02"? Is it the area covered by domain 2? In case 2-way nesting is applied, you will see the results of D03 in D02. I also think that you cannot assume that in case the results in D03 are reasonable, it's the same in D02.

Page 10, lines 21-27: I think this should be put behind the paragraph explaining the formulas. 24 h sums are not sub-daily anymore.

Page 11, first paragraph: What is the "tested value"? Is it the forecast model? In your formulas, the total number of time steps should be 2,4,6,8 as you are using 3 hourly precipitation data. Did you consider this in your evaluation?

Page 11, line 11: It is still not clear to me, how the maximum errors are defined. Are they based on 6h, 12h, 18h, or 24 h precipitation differences? Is it the same value for all experiments? It should be reasonable to use a single value for the different time periods and spin-up times. In general, it is very confusing to read and interpret the rescaled scores. Did you choose the scores

only to fit all scores into one figure? Sikder and Hossain (2016) had a different intention behind. What is R' (not explained)?

Page 11, lines 21-22: This is already mentioned before and can be deleted.

Page 11, lines 23-24: Is it the same domain or the same area? This makes a difference.

Page 12, line 29: Are you interested in D02 or D03? Are you evaluating on the area of D03, even if you are talking about D02? "scale of D03" is strongly misleading here.

Page 13, line 10: What is a "PW-related" feature? This is unclear.

Page 13, line 26: I do not think that there is a really obvious variation in R' here.

Page 13, line 28-29: How can this be justified? Please explain.

Page 14, line 13: This is hardly visible for me in case of WR'.

Page 14, lines 13-15: What do you mean by surface perturbations? Of course, the coarse resolution of the initial conditions severely limits the meaningfulness.

Page 14, line 25: I do not see this in your plots.

Page 15, line 16: What is meant by a "diurnal tendency"?

Page 15, lines 27-29: Bias is commonly used when showing absolute differences.

Page 15, line 30: Not necessarily true.

Page 16, first paragraph: The data sets you applied for verification of precipitation are different in Fig. 7 and Fig. 8 (CMORPH merged vs. CMC).

Page 17, line 17: Avoid "excessively". This really gives a negative touch to the application of large domains and very high resolution.

Page 17, line 9: It's quite confusing for the reader that an increasing RMSE is good.

Page 17, line 14: What do you mean here? Please be more precise.

Page 17, line 21: This a major point. In my opinion you cannot expect that you are able to reproduce a single extreme event in case you apply very coarse initial and boundary conditions. There is a high chance, that applying data assimilation is essential here. See e.g. Sun et al. (2013) MWR. I do not think that on a short time period, the boundary conditions take the leading role.

Page 18, line 1: What kind of regional geographical data sets did you apply? Did you provide your own landuse and/or soil texture data set? Did you try this data set: https://cera-www.dkrz.de/WDCC/ui/cerasearch/entry?acronym=WRF_NOAH_HWSD_world_TOP_SOILTYP ? I am not sure if the default coarse FAO data set is sufficient here.

Page 18, line 19: Where does the cumulative spatial error come from? How is it calculated? Please mention this in the manuscript.

*Figures*: Please use panel subscripts when applicable.

Figure 1: Please rework on the terrain plot. The default NCL color bar is not appropriate here. The image is either saturated in blue or red.

Figure 2: This image is overloaded. What is the meaning of the blue box?

Figure 4: The label bar in its current stage is useless. Why is such a coarse interval chosen? The reader does not see any major differences here. I would also appreciate a color plot here. For readers who are not familiar with China, please indicate where Beijing is located. Please rewrite the figure caption. This is very hard to understand.

Figure 6: In the 24 h diagram, WRMSE' of C7 appears to be missing.

*Tables*

Table 1: Is the threshold value the same for all time periods (see my comment above)?

Table 2: Are the results based on the area of domain 3?

---

## Author Response (AR2)

Dear Associate Editor/Reviewers,

We would like to thank the editor and two anonymous referees for the time spent on reviewing our manuscript and the comments helping us improving the article. We have made corrections according to these comments and marked the amended portion in blue in the annotated manuscript. Below, we provide our point-to-point response to the comments and present the main corrections in the revised manuscript.

**Response to the reviewer's comments**

"The authors present a study to investigate the impact of different domain sizes, vertical resolution, nesting ratios and spin-up time on a heavy precipitation event over Beijing. The simulations were forced by ERA-Interim reanalysis data available on 0.75° resolution in six hourly intervals. The different experiments were performed using three domains with a nesting approach while the innermost domain centered on Beijing. Sub-daily precipitation of the third domain was verified against a merged precipitation product in 3 hourly intervals.

Generally, this type of study is of importance as domain size, vertical and horizontal resolution can either have a positive or detrimental impact on the forecast quality. Compared to the first draft, the English spelling and grammar is considerably improved."

**Response**: Thank you very much for the encouraging feedback.

**General Comments**

**1.** - For me the major concern of the study is to investigate a sub-daily heavy rainfall event and drive the model with coarse resolution ERA-Interim data. I recommend repeating at least one of the simulations by applying the high-resolution ECMWF operational analysis or ERA5 reanalysis data.

**Response**: We really appreciate your suggestion. We agree that the quality of the forcing data is one factor that could affect the accuracy of regional heavy rainfall predictions. But in this case, our main focus is on evaluating the possible effects of the WRF domain configurations and the spin-up time at sub-daily and convective-resolving scales and emphasizing the importance of reevaluation of these model configuration aspects in regional SDHR-related applications. In addition, given that ERA5 is just available from March 2018, we think that another thorough study (in progress) is needed on evaluating this high-resolution dataset.

**2.** - I also feel that the rescaled scores cause more confusion to the reader, rather than improving the readability.

**Response**: Thank you for pointing it out. The metrics are adjusted to ensure that the ideal value of all of the metrics is 1, which has been proven to be helpful for evaluation. We have added some missing information to increase the readability of the manuscript. **Please see Page 11, line 7-18.**

**Specific comments**

**1.** – From the manuscript, it is not clear to me, whether you applied one-way or two-way nesting. This information is missing for the reader.

**Response**: Thank you for pointing it out. We use two-way nesting approach in this study. The missing information has been added in the revised manuscript. **Please see Page 8, line 27.**

**2.** – The size of your innermost domain is never mentioned but this can play an important role. In case it's just $100 \times 100$ cells, you have to subtract 10(5) cells in both direction due to boundary effects so you will have an effective grid size of 90 $\times$ 90 levels and I have a feeling that this is not sufficient at this particular high resolution.

**Response**: Thank you for pointing it out. The grid numbers are $90 \times 90$ in Case 0, $250 \times 250$ in Case 11, and $490 \times 490$ in Case 12. We have added this missing information in the revised manuscript. **Please see Page 9, line 4 and Table 1 on Page 34**.

**3.** – For a better readability, I suggest the following section ordering:

Section 2 Meteorological situation: The paragraph from section 3.1 on page 7 can be moved here.

Section 3 Experimental design

3.1 WRF Model: The paragraph on page 5, Line 7-17 can be moved here.

3.2 Model configuration

- Forcing: The paragraph on page 4, starting line 24 can be moved here.

- Model setup end experiments: The text between page 8, lines 5 to page 9, line 29 can be moved here.

**Response**: Thank you for your suggestions. We have moved the first paragraph in Section 2 to Section 3.1. In addition, the subtitle of 3.1 has been amended to "Study Event Selection and WRF Physical Schemes" to increase the readability of the manuscript. **Please see Page 6, line 20 and Page 7, line 21-32.**

**4**. – Abstract: What do you mean with "cumulative spatial error"? I do not find any explanation in the manuscript.

**Response**: Thank you for pointing it out. We have amended it to "the root mean squared error". **Please see Page 1, line 21**.

**5**. – Page 2, line 24: Which period?

**Response**: Thank you for raising this question. Here it refers to the time when substantial improvements in the predictive skill of NWP were made at the end of the 20th century. To make it clear, the original statement has been amended to "The ice wasn't broken until the end of the 20th century…The NWP models developed during and after this period can perform regional and convective-scale modelling and display good performance in simulating heavy rainfall." **Please see Page 2, line 21-25.**

**6**. – Page 2, line 25: Please use Skamarock et al. (2008) as reference here.

**Response**: Thank you for your suggestion. We have added the reference as you suggested. **Please see Page 2, line 26**.

**7**. – Page 3, line 18: "relatively" repeated twice. I think "relative" is not appropriate here.

**Response**: Thank you for pointing it out. This sentence has been amended to "However, these aspects of model configuration have received **less** attention in regional case studies because of **their insignificant effects** on rainfall forecasts in coarse-resolution and long-term model simulations when compared to the physics of the WRF model." **Please see Page 3, line 17-19**.

**8**. – Page 3, line 21: I do not agree here. For sure there are studies about this topic.

**Response**: Thank you for raising this question. Here, we use "**Generally**" to express that in most regional case studies, the model configuration aspects, such as the spatial resolutions and the spin-up time, are left at the **common settings** recommended by the official website of the WRF model and by some experimental regional heavy rainfall studies. **Please see Page 3, line 20-22.**

**9**. – Page 3, line 25: I guess you are talking about Beijing here.

**Response**: Thank you for pointing it out. We have added "happened in Beijing" after "a regional SDHR event". **Please see Page 3, line 26**.

**10**. – Page 4, line 4: Did you use the setting from the two publications in your study? If this is the case, please mention it.

**Response**: Thank you for raising this question. To make it clear, the sentence has been replaced by "several convective-scale studies have been carried out to re-evaluate the optimal combination of the physics options used in the WRF model, such as Di et al. (2015) and Wang et al. (2015). These studies represent background information that stimulates this research." **Please see Page 4, line 3-5.**

**11**. – Page 4, line 6: What is the first question here?

**Response**: The first question we attempt to explore is "whether the recommended configuration of WRF represents the best choice in reproducing the Beijing SDHR heavy rainfall event", which is mentioned **on Page 3, line 24 to 26**.

**12**. – Page 4, line 15: Reference to ERA-Interim?

**Response**: Thank you for pointing it out. We have added Dee et al. (2011) after "ERA-Interim". **Please see Page 4, line 16.**

**13**. – Page 5, line 1: I think Dee et al. (2011) are sufficient here.

**Response**: Thank you for your suggestion. Given that Berrisford et al. (2009) also provides a detailed description of the ERA-Interim product, we think it is appropriate to add this reference here.

**14**. – Page 5, lines 4-5: What is the CMC model saying in this case? Is it better than WRF? I am also not confident that the CMC model configuration is the justification to apply coarse resolution input and boundary conditions.

**Response**: Thank you for raising this question. It is well known that the height of the atmospheric layer can change with the variations of seasons and geographic locations. Therefore, although ERA-Interim provides forcing data at the same pressure level, the vertical resolution of the dataset differs across different regions. That's the reason why we present a table showing the approximate height at each pressure level over Beijing during the summertime, as well as saying the vertical levels used by the CMC models for regional forecasting. We agree that the accuracy of the initial dataset could affect the predictive skills of the WRF model. In the revised version, we have added one experiment forced by the model level data for comparison and have discussed it in Section 6. **Please see Page 17, line 26-28.**

**15**. – Page 5, lines 19-29: Please check if the full paragraph is necessary here. The focus in your study is not evaluating LBCs.

**Response**: Thank you for pointing it out. In this paragraph, the main concern is to describe the effects of WRF domain size configuration and to present the general guidance on selecting the appropriate domain size, not LBCs.

**16**. – Page 5, line 31: Do you really mean grid spacing here? Or do you mean the number of grid cells.

**Response**: Thank you for raising this question. We mean grid spacing here.

**17**. – Page 6, line 1-2: I think a vertical resolution of > 1 km in the PBL is far not sufficient here. I also do not agree with this statement in general.

**Response**: Thank you for pointing it out. It is well known that the rainfall processes mainly happen in the lower and middle troposphere, and the vertical turbulence features in the PBL layer could affect the generation of rainfall. Therefore, to well represent the rainfall process, the vertical resolution should differ across different layers, often with the highest vertical resolution in the PBL layer. Here the vertical resolution of 1 km refers to the mean value that is averaged among all of the vertical distances between the adjacent vertical layers.

**18**. – Page 6, line 7: What is excessive grid spacing?

**Response**: Thank you for pointing it out. We have amended it to "too small horizontal grid spacings." **Please see Page 5, line 27**.

**19**. – Page 6, line 10: see my comment two lines above. I think this is wrong.

**Response**: Please see the replies above. The vertical resolution of 1 km refers to the mean value that is averaged among all of the vertical distances between the adjacent vertical layers.

**20**. – Page 6, line 18-20: Do you really mean spin-up time here? Or do you mean something like forecast lead times?

**Response**: Thank you for raising this question. We have amended it to "forecast lead times." **Please see Page 6, line 5-8.**

**21**. – Page 6, line 25: Are you sure?

**Response**: Thank you for raising this question. The original statement has been replaced by "however, this spin-up time is often regarded as the suitable choice in many regional case studies without further verification." **Please see Page 6, line 12-13.**

**22**. – Page 7, line 15: I would rather write 60 m to 2300 m.

**Response**: Thank you for your suggestion. We have adjusted this statement as you suggested. **Please see Page 7, line 2**.

**23**. – Page 7, line 22: You mention 100 mm/h rain rate, but your verification is performed over at least 6 hourly windows. How does this fit together?

**Response**: Thank you for raising this question. The observed maximum rainfall intensity of 100 mm/h is only mentioned when describing the characteristics of the study event, which has nothing to do with the verification processes.

**24**. – Page 8, line 5: What is the reason you also apply the cumulus parameterization in D03. Please explain. I am not sure if the GD is designed to run at very high resolution.

**Response**: Thank you for raising this question. In this study, GD cumulus parameterization is turned on for each domain, including D03, to represent the effects of sub-grid scale convective processes which are also detected in the rainfall processes of this heavy rainfall event.

**25**. – Page 8, line 10: I think Ek et al. is the wrong reference for the MM5 surface layer scheme.

**Response**: Thank you for pointing it out. We have checked the full name of the surface layer scheme used in this study. That is the "Monin-Obukhov surface layer scheme" that proposed by Ek et al. (2003). We have added the missing information in the revised manuscript. **Please see Page 8, line 8-9**.

**26**. – Page 8, lines 22-25: I think this is not correct as you also change the spin-up period. There is no other choice than using the same map projection in a multi domain simulation.

**Response**: Thank you for pointing it out. To make it clear, we have adjusted the sentences to "The initial datasets and the model physics are the same for all of the domains throughout the entire comparative procedure. Because the area of interest is located in the middle latitudes, the Lambert conformal projection is employed in all of the experiments, which is centred on the same latitude (42.25° N) and longitude (114.0° E)." **Please see Page 8, line 21-24**.

**27**. – Page 8, line 27: Please mention the recommended setting you refer to in section 2. What is the time step you applied? This is not mentioned in the manuscript? If an adaptive time step is applied, then the results are not comparable.

**Response**: Thank you for pointing it out. We apply the time step of 90 s for most experiments except the Case 4 and Case 7. Instability was encountered while running these two cases and the model ran much slower than we expected and stopped before the end time.

**28**. – Page 8, line 29: The odd ratio is selected due to the applied Arakawa grid.

**Response**: Thank you for pointing it out. The original statement has been replaced by "An odd downscaling ratio (1:3:3) is selected to reduce the initial error introduced by interpolating the initial fields to the assigned Arakawa grid." **Please see Page 8, line 28-29**.

29. – Page 9, lines 2-3: This should be clear.

**Response**: Thank you for pointing it out. We have added the missing information "The grid numbers of D01, D02 and D03 are 40×40, 72×72 and 90×90, respectively" in the revised version. **Please see Page 9, line 4**.

**30**. – Page 9, line 6: Why did you output the D03 data only in 3 hourly intervals? Is it due to the merged CMORPH data set? Why didn't you use the hourly gridded precipitation data set from the CMC? Especially when you are interested in sub-daily extreme rainfall?

**Response**: Thank you for raising this question. The reason that the 0.05-degree merged CMORPH dataset is chosen as the reference data because it has finer spatial resolution than the 0.1-degree CMC dataset, which is more suitable for convective-scale verifications.

**31**. – Page 9, lines 11-13: This implies that your simulation C0 is unusable here as it does not capture the situation. What is a perturbed synoptic feature?

**Response**: Thank you for raising this question. To make it clear, we have amended this statement to "For computational efficiency, the MCS systems that drive **the local synoptic features** are not completely contained within the outermost domain of C0, the information of which is compensated by the updated LBCs from ERA-Interim." **Please see Page 9, line 11-13**.

**32**. – Page 9, line 16: This is totally confusing. D02 of the experiment C0 (C2) is now the new D01 for C1? I do not see the equivalence here: 72×72 cells at 13.5 km vs. 80×64 cells at 40.5 km resolution.

**Response**: There are 80×64 cells at 40.5 km in D01 of C1 and 240×192 cells at 13.5 km in D02 of C2. The equivalence is also seen as follows: 40×40 cells at 40.5 km in D01 of C0 vs. 120×120 cells at 13.5 km in D02 of C1.

**33**. – Page 9, line 22: I guess you mean the middle troposphere. Why is this condition not satisfied in other regions?

**Response**: Thank you for raising this question. One thing that we would like to make it clear is that the height of the atmospheric layer can change with the variations of seasons and geological locations. Therefore, the vertical resolution of the ERA-Interim pressure level data may differ over different regions.

**34**. – Page 9, line 25: I think there is no "increased nesting ratio" of 1:3:3.

**Response**: Thank you for pointing it out. We have amended the statement to "with the increased nesting ratio of 1:3:3, 1:5:5 and 1:7:7." **Please see Page 9, line 24-26.**

**35**. – Page 10, line 6: ERA-Interim provides precipitation on a 6 hourly basis.

**Response**: ERA-Interim reanalysis only provides precipitation on a daily basis.

**36**. – Page 10, line 12: What do you mean with "scale of D02"? Is it the area covered by domain two?

**Response**: Thank you for raising this question. To make it clear, we have amended it to "over domain two". **Please see Page 10, line 10**.

**37**. – Page 10, line 30: 24 h sums are not sub-daily anymore.

**Response**: Thank you for raising this question. Here the 24 h results are not calculated based on the 24 h sums but on the mean value averaged over each time step, which has been mentioned in the manuscript **on Page 10, line 24-26**.

**38**. – Page 11, first paragraph: What is the "tested value (field)"? Is it the forecast model (outputs)? In your formulas, the total number of time steps should be 2, 4, 6, 8 as you are using 3 hourly precipitation data. Did you consider this in your evaluation?

**Response**: Thank you for pointing it out. Here, the tested value refers to the model outputs. The confusing sentence on **Page 11, line 3** has been replaced by "N is the total number of time steps, depending on the time period considered."

**39**. – Page 11, line 11: It is not clear to me, how the maximum errors are defined. Are they based on 6h, 12h, 18h, or 24h precipitation differences? Is it the same value for all experiments? It should be reasonable to use a single value for the different time periods and spin-up times. In general, it is very confusing to read and interpret the rescaled scores. Did you choose the scores only to fit all scores into one figure? What is R' (not explained)?

**Response**: Thank you for raising this question. The factor used for rescaling is determined by the largest values of each error metric in all of the experiments and keeps the same for all of the evaluated time periods. The metrics are adjusted to ensure that the ideal value of all of the metrics is 1, with the purpose of facilitating evaluation. R' is the Pearson correlation coefficient, which has the same meaning and values as R. The correlation between R' and R can be seen in Table 2. The missing information has been added in the revised manuscript, **Please see Page 11, lines 10-11 and line 14**.

**40**. – Page 11, line 21-22: This is already mentioned before and can be deleted.

**Response**: Thank you for your suggestion. We have deleted this statement as you suggested.

**41**. – Page 11, line 23-24: Is it the same domain or the same area? This makes a difference.

**Response**: Thank you for raising this question. We have amended it to "over domain two (D02)" to make it clear. **Please see Page 11, line 25**.

**42**. – Page 12, line 29: Are you interested in D02 or D03? Are you evaluating on the area of D03, even if you are talking about D02? "scale of D03" is strongly misleading here.

**Response**: Thank you for raising this question. The comparison over the D02 area is used only as an auxiliary method for subjective verification when apparent discrepancies are noted in the results obtained for the inner domain (D03) and the outer domain (D02). **Please see Page 10, line 9-13**.

**43**. – Page 13, line 10: What is a "PW-related" feature? This is unclear.

**Response**: Thank you for raising this question. We have added the related information **on Page 10, line 22**.

**44**. – Page 13, line 26: I do not think that there is a really obvious variation in R' here.

**Response**: Thank you for raising this question. The statement is "The most obvious difference from the domain size scenario is that the values of R' calculated between the simulations and the ground truth vary slightly and remain almost the same between the different time periods." Here, it means that R' is detected with small variations.

**45**. – Page 13, line 28-29: How can this be justified? Please explain.

**Response**: Thank you for your question. In this study, $WRMSE'$ and $WR'$ are employed in identifying departures of the WRF simulations from the driving ERA-Interim weather fields as the model setup is varied. The reason has been mentioned in Section 1. **Please see Page 4, line 16-17**.

**46**. – Page 14, line 13: This is hardly visible for me in case of WR'.

**Response**: Thank you for pointing it out. The original statement has been replaced by "Examining the values of $WRMSE'$ and $WR'$ shows that the differences between the simulations and the reanalysis are more distinct in C3 and C4 than in C1." **Please see Page 14, line 12-13**.

**47**. – Page 14, line 13-15: What do you mean by surface perturbations? Of course, the coarse resolution of the initial conditions severely limits the meaningfulness.

**Response**: Thank you for raising this question. The surface perturbations include the small turbulence features that could be transported vertically. We agree that finer initial forcing data could be helpful to improve the NWP predictive skills, but whether or not the accuracy of the rainfall predictions can be enhanced also depends on the development level of the PBL and Land surface model.

**48**. – Page 14, line 25: I do not see this in your plots.

**Response**: Thank you for pointing it out. We have deleted the confusing statement.

**49**. – Page 15, line 16: What is meant by a "diurnal tendency"?

**Response**: Thank you for raising this question. It can be seen from Fig. 7 that the performance of the model runs show regular changes before 72h, with better performance detected in the model runs that initialized during daytime and worse performance identified in the model runs initialized during nighttime.

**50**. – Page 15, line 27-29: Bias is commonly used when showing absolute differences.

**Response**: Thank you for pointing it out. It is the absolute differences here.

**51**. – Page 15, line 30: Not necessarily true.

**Response**: Thank you for pointing it out. We have adjusted our statement to "This result may occur because the atmospheric water vapour content determines the maximum possible rainfall amount." **Please see Page 15, line 28-29**.

**52.** – Page 17, line 17: Avoid "excessively". This really gives a negative touch to the application of large domains and very high resolution.

**Response**: Thank you for your suggestion. The original statement has been replaced by "Moreover, experiments with too large domains, too high spatial resolutions, or too long spin-up times also yield poor performance in rainfall simulations." **Please see Page 16, line 15-16**.

**53**. – Page 17, line 9: It's quite confusing for the reader that an increasing RMSE is good.

**Response**: Thank you for pointing it out. We have amended this statement to "Specifically, $R'$ increases from 0.226 in C0 to 0.67 in C12; $RMSE'$ increases from 0.098 to 0.402; and $PMAX'$ increases from 0.44 to 0.883." **Please see Page 17, line 8-9**.

**54**. – Page 17, line 14: What do you mean here? Please be more precise.

**Response**: Thank you for pointing it out. To make it clear, the sentence has been revised to "The use of different time periods helps to determine the optimal configurations with higher physical rationality, such as the selection of the proper domain size." **Please see Page 17, line 15-17**.

**55**. – Page 17, line 21: There is a high chance, that applying data assimilation is essential here. See e.g. Sun et al. (2013) MWR. I do not think that on a short time period, the boundary conditions take the leading role.

**Response**: Thank you for pointing it out. We agree that the use of data assimilation methods has been shown to enable the extension of the lead time to 24 hours. However, this lead time is still insufficient to provide effective flood mitigation for medium or large urban areas with very short hydrologic response times.

**56**. – Page 18, line 1: What kind of regional geographical data sets did you apply? Did you provide your own landuse and/or soil texture dataset?

**Response**: Thank you for raising this question. We used 30-second static geographical data downloaded from the WRF official website to initialize the surface fields of the WRF model, which we have mentioned on **Page 7, line 20**.

**57**. – Page 18, line 19: Where does the cumulative spatial error come from? How is it calculated? Please mention this in the manuscript.

**Response**: Thank you for raising this question. We have amended it to "*RMSE*". **Please see Page 18, line 18**.

**58**. – Figure 2: This image is overloaded. What is the meaning of the blue box?

**Response**: Thank you for raising this question. We use the blue box to mark the main synoptic features that triggered the Beijing SDHR event to illustrate that ERA-Interim reanalysis captures the vortex (detected in 700pha) and the subtropical high pressure (detected in 500pha) well that occurred at the beginning of the rainfall event.

**59**. – Figure 4: The label bar in its current stage is useless. Why is such a coarse interval chosen? The reader does not see any major differences here. I would also appreciate a color plot here. For readers who are not familiar with China, please indicate where Beijing is located. Please rewrite the figure caption. This is very hard to understand.

**Response**: Thank you for your suggestion. To make it clear, we added the 6-h spatial distribution of the accumulated precipitation during the fourth 6h durations in Fig. 4, from which the differences between the three experiments could be seen more evident. The boundary line of the Beijing Area in Fig. 4 is the same as that shown in Fig.1. Besides, the figure caption has been rewritten as you suggested. **Please see Page 29**.

**60**. – Figure 6: In the 24 h diagram, WRMSE' of C7 appears to be missing.

**Response**: Thank you for pointing it out. We have added the missing value in Fig. 6. **Please see Page 31**.

We really appreciate your help in improving this manuscript, and we hope that our replies have addressed your concerns.

Kind Regards,

The Authors

**List of all relevant changes**

**Page 1, Line 21** "cumulative spatial error" was replaced by "the root mean squared error".

The correction was made based on the Specific Comment 4 of Reviewer #1.

**Page 2, after Line 21** the original statements were replaced by "The ice wasn't broken until the end of the 20th century; substantial improvements in the predictive skill of NWP were made that resulted from the increases in computational power and storage capacity, which enable parallel processing of high-resolution forcing data and the resolution of convective-scale physical processes (Done et al., 2004; Clark et al., 2016). The NWP models developed during and after this period can perform regional and convective-scale modelling and display good performance in simulating heavy rainfall."

The correction was made based on the Specific Comment 5 of Reviewer #1.

**Page 2, Line 25** "Skamarock et al. (2008)" was added after "WRF model".

The correction was made based on the Specific Comment 6 of Reviewer #1.

**Page 3, after Line 17** the original statement was replaced by "However, these aspects of model configuration have received less attention in regional case studies because of their insignificant effects on rainfall forecasts in coarse-resolution and long-term model simulations when compared to the physics of the WRF model."

The correction was made based on the Specific Comment 7 of Reviewer #1.

**Page 3, after Line 20** the original statement was replaced by "Generally, these model configuration aspects are left at the common settings recommended by the official website of the WRF model and by some experimental regional heavy rainfall studies."

**Page 3, Line 26** "happened in Beijing" was added after "a regional SDHR event".

The correction was made based on the Specific Comment 9 of Reviewer #1.

**Page 4, after Line 3** the original statement was replaced by "several convective-scale studies have been carried out to re-evaluate the optimal combination of the physics options used in the WRF model, such as Di et al. (2015) and Wang et al. (2015). These studies represent background information that stimulates this research."

The correction was made based on the Specific Comment 10 of Reviewer #1.

**Page 4, Line 16** "(Dee et al., 2011)" was added after "ERA-Interim".

The correction was made based on the Specific Comment 12 of Reviewer #1.

**Page 5, Line 27** "excessive grid spacings" was replaced by "too small grid spacings".

The correction was made based on the Specific Comment 18 of Reviewer #1.

**Page 6, after Line 5** the original statements were replaced by "Therefore, in cases where short forecast lead times are expected, e.g., real-time rainfall forecasting, the spin-up time is mainly determined by the domain size and the regional initial and boundary conditions. However, in cases where long forecast lead times are needed, e.g., warnings of extreme rainfall, the effects of chaotic behaviour should be relatively evident."

The correction was made based on the Specific Comment 20 of Reviewer #1.

**Page 6, after Line 12** the original statement was replaced by "however, this spin-up time is often regarded as the suitable choice in many regional case studies without further verification"

The correction was made based on the Specific Comment 21 of Reviewer #1.

**Page 6, Line 20** "WRF Model Configuration" was replaced by "WRF Physical Schemes".

**Page 7, Line 2** "2 300 m to 60 m" was replaced by "60 m to 2 300 m".

The correction was made based on the Specific Comment 22 of Reviewer #1.

**Page 8, after Line 8** the original statement was replaced by "The Noah land-surface model (Chen and Dudhia, 2001) is used and coupled with the Monin-Obukhov surface layer model (Ek et al., 2003)."

**Page 8, after Line 21** the original statements were replaced by "The initial datasets and the model physics are the same for all of the domains throughout the entire comparative procedure. Because the area of interest is located in the middle latitudes, the Lambert conformal projection is employed in all of the experiments, which is centred on the same latitude (42.25° N) and longitude (114.0° E)."

The correction was made based on the Specific Comment 26 of Reviewer #1.

**Page 8, Line 27** "two-way" was added before "nested domains".

The correction was made based on the Specific Comment 1 of Reviewer #1.

**Page 8, after Line 28** the original statement was replaced by "An odd downscaling ratio (1:3:3) is selected to reduce the initial error introduced by interpolating the initial fields to the assigned Arakawa grid."

**Page 9, Line 4** we added:

"The grid numbers of D01, D02, and D03 are $40\times40$, $72\times72$ and $90\times90$, respectively."

The correction was made based on the Specific Comment 2 of Reviewer #1.

**Page 9, after Line 11** the original statement was replaced by "For computational efficiency, the MCS systems that drive the local synoptic features are not completely contained within the outermost domain of C0, the information of which is compensated by the updated LBCs from ERA-Interim."

**Page 9, after Line 24** the original statements were replaced by "The three experiments (OS2, C6, and C7) in scenario three (S3) differ in terms of their horizontal resolutions and nesting ratios, with increased nesting ratio of 1:3:3 (4.5 km grid spacing in D03), 1:5:5 (1.62 km in D03) and 1:7:7 (0.826 km in D03)."

The correction was made based on the Specific Comment 34 of Reviewer #1.

**Page 10, Line 10** "on scale of D02" was replaced by "over domain two".

The correction was made based on the Specific Comment 36 of Reviewer #1.

**Page 10, Line 22** "(PW-related metrics)" was added after "The two metrics selected for the verification of PW".

The correction was made based on the Specific Comment 43 of Reviewer #1.

**Page 11, Line 3** the original statement was replaced by "$N$ is the total number of time steps, depending on the time period considered."

The correction was made based on the Specific Comment 38 of Reviewer #1.

**Page 11, after Line 10** the original statement was replaced by "The factor used for rescaling is determined by the largest values of each error metric in all of the experiments and keeps the same for all of the evaluated time periods (Sikder and

Hossain, 2016)."

The correction was made based on the Specific Comment 39 of Reviewer #1.

**Page 11, after Line 11** we added:

"$RE_{PMAX}$ and $RE_{TP}$ are added by 1 to have the ideal value of 1. The rescaled metrics are $PMAX'$ and $TP'$, respectively."

**Page 11, Line 14** we added:

"For example, $POD$ is replaced with $POD'$, and $R$ is replaced with $R'$. "

**Page 11, Line 25** "a slightly larger area" was replaced by "over domain two (D02)".

The correction was made based on the Specific Comment 41 of Reviewer #1.

**Page 14, after Line 12** the original statement was replaced by "Examining the values of $WRMSE'$ and $WR'$ shows that the differences between the simulations and the reanalysis are more distinct in C3 and C4 than in C1."

The correction was made based on the Specific Comment 46 of Reviewer #1.

**Page 15, after Line 28** the original statement was replaced by "Moreover, experiments with too large domains, too high spatial resolutions, or too long spin-up times also yield poor performance in rainfall simulations."

The correction was made based on the Specific Comment 51 of Reviewer #1.

**Page 16, after Line 15** the original statement was replaced by "This result may occur because the atmospheric water vapour content determines the maximum possible rainfall amount."

The correction was made based on the Specific Comment 52 of Reviewer #1.

**Page 17, after Line 8** the original statement was replaced by "Specifically, $R'$ increases from 0.226 in C0 to 0.67 in C12;

$RMSE'$ increases from 0.098 to 0.402; and $PMAX'$ increases from 0.44 to 0.883."

**Page 17, after Line 15** the original statement was replaced by "The use of different time periods helps to determine the optimal configurations with higher physical rationality, such as the selection of the proper domain size."

The correction was made based on the Specific Comment 54 of Reviewer #1.

**Page 18, Line 18** "the cumulative spatial error" was replaced by "$RMSE$".

The correction was made based on the Specific Comment 57 of Reviewer #1.

**Figures were amended:**

In Fig. 4, we added four subfigures showing the 6-h spatial distribution of the accumulated precipitation during the fourth

6h durations.

In Fig. 4, the missing value of $WRMSE'$ in the 24h diagram was added.

**Data in the tables were amended:**

The grid numbers of D03 was added in Table 1.

[revised manuscript text omitted]